# DSEG-LIME: Improving Image Explanation by Hierarchical Data-Driven Segmentation

## Abstract

Explainable Artificial Intelligence (XAI) is crucial in unravelling decision-making processes in complex machine-learning models. LIME (Local Interpretable Model-agnostic Explanations) is a well-known XAI framework for image analysis. It utilizes image segmentation to create features to identify relevant areas for classification. Consequently, poor segmentation can compromise the consistency of the *explanation* and undermine the importance of the segments, affecting the overall *interpretability*. To address these challenges, we introduce **DSEG-LIME** (Data-Driven Segmentation LIME), featuring: *i*) a *data-driven* segmentation for *human-recognized* feature generation by *foundation model* integration, and *ii*) a user steered granularity in the *hierarchical segmentation* procedure through *composition*. We evaluate DSEG-LIME on pre-trained models using ImageNet classes, explicitly targeting scenarios without domain-specific knowledge. Our findings demonstrate that DSEG outperforms most of the XAI metrics and enhances the alignment of explanations with human-recognized concepts, significantly improving interpretability.

## 1 Introduction

**Why should we trust you?** The integration of AI-powered services into everyday scenarios, with or without the need for specific domain knowledge, is becoming increasingly common. For instance, consider AI-driven systems that assist in diagnosing diseases based on medical imaging. In such high-stakes scenarios, accuracy and alignment with expert knowledge are paramount. To ensure reliability, stakeholders, including medical professionals and regulators, frequently seek to evaluate the AI's performance post-deployment. For example, one might assess whether the AI correctly identifies anomalies in medical scans that could indicate early-stage cancer. The derived question - "Why should we *trust* the model?" - directly ties into the utility of *Local Interpretable Model-agnostic Explanations* (LIME) (Ribeiro et al., 2016). LIME seeks to demystify AI decision-making by identifying key features that influence the output of a model, underlying the importance of the Explainable AI (XAI) research domain, particularly when deploying opaque models in real-world scenarios (Barredo Arrieta et al., 2020; Linardatos et al., 2021; Garreau & Mardaoui, 2021).

**Segmentation is key.** LIME uses segmentation techniques to identify and generate features to determine the key areas of an image that are critical for classification. However, a challenge emerges when these segmentation methods highlight features that fail to align with identifiable, clear concepts or arbitrarily represent them. This issue is particularly prevalent with conventional segmentation techniques. These methods, often grounded in graph- or clustering-based approaches (Wang et al., 2017), were not initially designed for distinguishing between different objects within images. However, they are the default in LIME's implementation (Ribeiro et al., 2016).

**Ambiguous explanations.** The composition of the segmentation has a significant influence on the explanation's *quality* (Schallner et al., 2020). Images with a large number of segments frequently experience significant stability issues in LIME, primarily due to the increased number of sampled instances (Section 2). This instability can lead to the generation of two entirely contradictory explanations for the same instance, undermining trust not only in LIME's explanations but also in the reliability of the model being analyzed (Garreau & Mardaoui, 2021; Alvarez-Melis & Jaakkola, 2018; Zhou et al., 2021; Zhao et al., 2020; Tan et al., 2024). Moreover, humans often struggle to

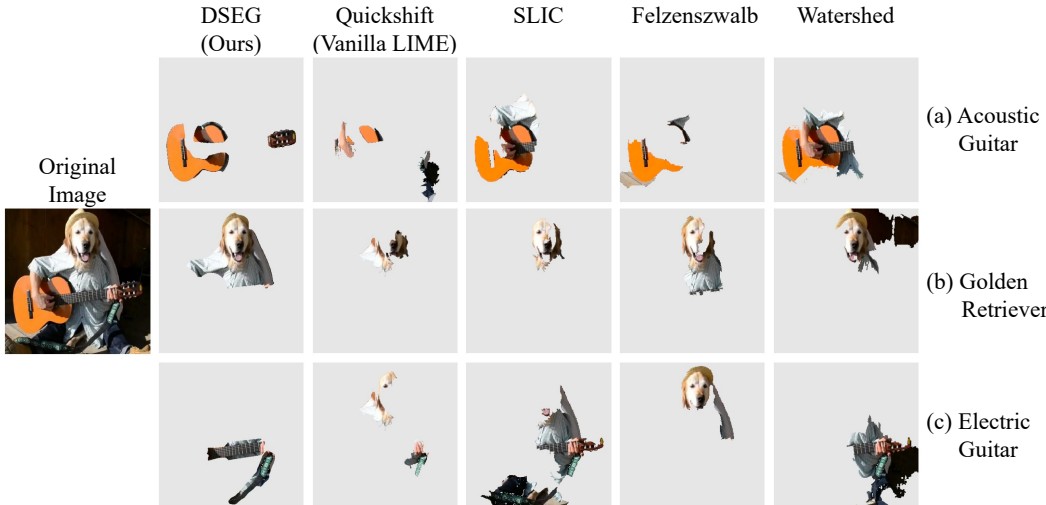

Figure 1: **Segmentation techniques within LIME.** We illustrate LIME-generated explanations (Ribeiro et al., 2016) for EfficientNetB4 (Tan & Le, 2019), utilizing various segmentation methods: DSEG (ours) combined with SAM (Segment Anything) (Kirillov et al., 2023), Quickshift (Hoyer et al., 2019), SLIC (Simple Linear Iterative Clustering) (Achanta et al., 2012), Felzenszwalb (Felzenszwalb & Huttenlocher, 2004), and the Watershed algorithm (Neubert & Protzel, 2014). The top predictions are 'Acoustic Guitar' ($p = 0.31$), 'Golden Retriever' ($p = 0.24$), and 'Electric Guitar' ($p = 0.07$). Among these, our method (DSEG) provides the clearest and most interpretable concept representations.

*interpret* the explanations, as the highlighted areas do not align with our intuitive understanding (Molnar et al., 2022; Kim et al., 2022).

**This work.** In this paper, we address the challenges above by introducing **DSEG-LIME** (**D**ata-driven **Seg**mentation **LIME**), an adaptation of the LIME framework for image analysis across domains where specialized knowledge is not required. We replace the conventional segmentation algorithm with *foundation models*, such as SAM (Kirillov et al., 2023), and often refer to these models as *data-driven* to emphasize their capability to generate features that more effectively capture human-recognizable concepts, leveraging insights derived from extensive image datasets. Given the great segmentation ability of such models, we implement a *compositional object structure*, adapting LIME's feature generation with a novel *hierarchical segmentation*. This adaptation provides flexibility in the granularity of concepts, allowing users to specify the detail of LIME's explanation, viewing a car as a whole or in parts like doors and windshields. This approach breaks down broad categorizations, enabling independent evaluation of each sub-concept. Figure 1 demonstrates the motivation mentioned above by employing LIME, which generates explanations using various segmentation techniques, specifically focusing on an image of a dog playing the guitar. In this context, DSEG excels by more clearly highlighting features that align with human-recognizable concepts, distinguishing it from other methods.

**Contribution.** The key contributions of our paper are summarized as follows: *(i)* We present DSEG-LIME, an enhanced version of the LIME framework for image analysis, leveraging foundation models to improve image segmentation. *(ii)* DSEG extends LIME by incorporating compositional object structures, enabling hierarchical segmentation that offers users adjustable feature granularity. *(iii)* We rigorously evaluate our approach with other segmentation methods and LIME enhancements across multiple pre-trained image classification models. Our evaluation includes a user study for qualitative insights and distinguishes between explaining (*quantitative*) and interpreting (*qualitative*) aspects. We acknowledge that explanations considered intuitive by users may not always reflect the AI model's operational logic, which can diverge from human perception (Molnar et al., 2022; Freiesleben & König, 2023). To address this, we complement our evaluation with several quantitative performance metrics widely used in XAI research (Nauta et al., 2023).

## 2 RELATED WORK

**Region-based perturbation XAI techniques.** LIME is among several techniques designed to explain black box models through image perturbation. Fong & Vedaldi (2017) introduced a meta-predictor framework that identifies critical regions via saliency maps. Subsequently, Fong et al. (2019) developed the concept of extremal perturbations to address previous methods' limitations. Additionally, Kapishnikov et al. (2019) advanced an integrated-gradient, region-based attribution approach for more precise model explanations. More recently, Escudero-Viñolo et al. (2023) have highlighted the constraints of perturbation-based explanations, advocating for the integration of semantic segmentation to enhance image interpretation.

**Instability of LIME.** The XAI community widely recognizes the instability in LIME's explanations, which stems from LIME's design (Alvarez-Melis & Jaakkola, 2018; Zhou et al., 2021; Zhao et al., 2020; Tan et al., 2024). Alvarez-Melis & Jaakkola (2018) handled this issue by showing the instability of various XAI techniques when slightly modifying the instance to be explained. A direct improvement is Stabilized-LIME (SLIME) proposed by Zhou et al. (2021) based on the central limit theorem to approximate the number of perturbations needed in the data sampling approach to guarantee improved explanation stability. Zhao et al. (2020) improved stability by exploiting prior knowledge and using Bayesian reasoning - BayLIME. GLIME (Tan et al., 2024) addressed this issue by employing an improved local and unbiased data sampling strategy, resulting in explanations with higher fidelity - similar to the work by Rashid et al. (2024). Recent advancements include Stabilized LIME for Consistent Explanations (SLICE) Bora et al. (2024), which improves LIME through a novel feature selection mechanism that removes spurious superpixels and introduces an adaptive perturbation approach for generating neighbourhood samples. Another hierarchical-based variation, DLIME Zafar & Khan (2021), utilizes agglomerative hierarchical clustering to organize training data, focusing primarily on tabular datasets. In contrast, DSEG-LIME extends this concept to images by leveraging the hierarchical structure of image segments.

**Segmentation influence on explanation.** The segmentation algorithm utilized to sample data around the instance $\mathbf{x}$ strongly influences its explanation. It directly affects the stability of LIME itself, as suggested by Ng et al. (2022). This behaviour is in line with the investigation by Schallner et al. (2020) that examined the influence of different segmentation techniques in the medical domain, showing that the quality of the explanation depends on the underlying feature generation process. Blücher et al. (2024) explored how occlusion and sampling strategies affect model explanations when integrated with segmentation techniques for XAI, including LRP (Layer-Wise Relevance Propagation) (Montavon et al., 2019) and SHAP (Lundberg & Lee, 2017). Their study highlights how different strategies provide unique explanations while evaluating the SAM technique in image segmentation. Sun et al. (2023) used SAM within the SHAP framework to provide conceptually driven explanations, which we discuss in Appendix B.4.

**Segmentation hierarchy.** The work of Li et al. (2022) aimed to simulate the way humans structure segments hierarchically and introduced a framework called Hierarchical Semantic Segmentation Networks (HSSN), which approaches segmentation through a pixel-wise multi-label classification task. HIPPIE (HIerarchical oPen-vocabulary, and unIvErsal segmentation), proposed by Wang et al. (2023), extended hierarchical segmentation by merging text and image data multimodally. It processes inputs through decoders to extract and then fuse visual and text features into enhanced representations.

## 3 FOUNDATIONS OF LIME

In this section, we introduce the LIME framework (Ribeiro et al., 2016), providing its theoretical foundation and functionality to establish the context for our approach.

**Notation.** We consider the scenario where we deal with imagery data. Let $\mathbf{x} \in \mathcal{X}$ represent an image within a set of images, and let $\mathbf{y} \in \mathcal{Y}$ denote its corresponding label in the output space with logits $\mathcal{Y} \subseteq \mathbb{R}$ indicating the labels in $\mathcal{Y}$. We denote the neural network we want to explain by $f : \mathcal{X} \to \mathcal{Y}$. This network functions by accepting an input $\mathbf{x}$ and producing an output in $\mathcal{Y}$, which signifies the probability $p$ of the instance being classified into a specific class.

### 3.1 LOCAL INTERPRETABLE MODEL-AGNOSTIC EXPLANATIONS

LIME is a prominent XAI framework designed to explain the decisions of a neural network $f$ in a *model-agnostic* and *instance-specific* (local) manner. It applies to various modalities, including images, text, and tabular data (Ribeiro et al., 2016). In the following, we will briefly review LIME's algorithm for treating images.

**Feature generation.** The technique involves training a local, interpretable surrogate model $g \in G$, where $G$ is a class of interpretable models, such as linear models or decision trees, which approximates $f$'s behavior around an instance $\mathbf{x}$ (Ribeiro et al., 2016). This instance needs to be transformed into a set of features that can be used by $g$ to compute the importance score of its features. In the domain of imagery data, segmentation algorithms segment $\mathbf{x}$ into a set of superpixels $s_0...s_d \in \mathcal{S}^D$, done by conventional techniques (Hoyer et al., 2019; Achanta et al., 2012; Felzenszwalb & Huttenlocher, 2004; Neubert & Protzel, 2014). We use these superpixels as the features for which we calculate their importance score. This step reflects the problematic process mentioned in Section 1, which forms the basis for the quality of the features that influence the explanatory quality of LIME.

**Sample generation.** For sample generation, the algorithm manipulates superpixels by toggling them randomly. Specifically, each superpixel $s_i$ is assigned a binary state, indicating this feature's visibility in a perturbed sample $\mathbf{z}$. The presence (1) or absence (0) of these features is represented in a binary vector $\mathbf{z}'_i$, where the $i$-th element corresponds to the state of the $i$-th superpixel in $\mathbf{z}$. When a feature $s_i$ is absent (i.e., $s_i = 0$), its pixel values in $\mathbf{z}$ are altered. This alteration typically involves replacing the original pixel values with a non-information holding value, such as the mean pixel value of the image or a predefined value (e.g., black pixels) (Ribeiro et al., 2016; Tan et al., 2024). Consequently, the modified instance $\mathbf{z}$, while retaining the overall structure of the original image $\mathbf{x}$, exhibits variations in its feature representation due to these alterations.

**Feature attribution.** LIME employs a proximity measure, denoted as $\pi_\mathbf{x}$, to assess the closeness between the predicted outputs $f(\mathbf{z})$ and $f(\mathbf{x})$, which is fundamental in assigning weights to the samples. In the standard implementation of LIME, the kernel $\pi_\mathbf{x}(\mathbf{z})$ is defined as follows:

$$\pi_\mathbf{x}(\mathbf{z}') = \exp\left(-\frac{D(\mathbf{x}', \mathbf{z}')^2}{\sigma^2}\right), \tag{1}$$

where $\mathbf{x}'$ is a binary vector, all states are set to 1, representing the original image $\mathbf{x}$. $D$ represents the $L2$ distance, given by $D(\mathbf{x}', \mathbf{z}') = \sqrt{\sum_{i=1}^{n}(\mathbf{x}'_i - \mathbf{z}'_i)^2}$ and $\sigma$ being the width of the kernel. Subsequently, LIME trains a linear model, minimizing the loss function $\mathcal{L}$, which is defined as:

$$\mathcal{L}(f, g, \pi_\mathbf{x}) = \sum_{\mathbf{z}, \mathbf{z}' \in \mathcal{Z}} \pi_\mathbf{x}(\mathbf{z}) \cdot (f(\mathbf{z}) - g(\mathbf{z}'))^2 \tag{2}$$

In this equation, $\mathbf{z}$, and $\mathbf{z}'$ are sampled instances from the perturbed dataset $\mathcal{Z}$, and $g$ is the interpretable model being learned (Ribeiro et al., 2016; Tan et al., 2024). The interpretability of the model is derived primarily from the coefficients of $g$. These coefficients quantify the influence of each feature on the model's prediction, with each coefficient's magnitude and direction (positive or negative) indicating the feature's relative importance and effect.

## 4 DSEG-LIME

In this section, we will present DSEG-LIME's two contributions: first, the substitution of traditional feature generation with a data-driven segmentation approach (Section 4.1), and second, the establishment of a hierarchical structure that organizes segments in a compositional manner (Section 4.2).

### 4.1 DATA-DRIVEN SEGMENTATION INTEGRATION

DSEG-LIME improves the LIME feature generation phase by incorporating data-driven segmentation models, outperforming conventional graph- or cluster-based segmentation techniques in creating recognizable image segments across various domains. Specifically, our approach mainly uses SAM (Segment Anything) (Kirillov et al., 2023) due to its remarkable capability to segment images in diverse areas. However, as the appendix shows, it can also be applied to other segmentation models. Figure 2 illustrates the integration of DSEG into the LIME framework, as outlined in

Section 3. Specifically, DSEG impacts the feature generation phase, influencing the creation of superpixels/features $\mathcal{S}^D$, and subsequently affecting the binary vector $\mathbf{z}'$ and the feature vector $\mathbf{z}$. This modification directly impacts the loss function in Equation (2) and the proximity metric for perturbed instances in Equation (1), leading to an improved approximation of the interpretable model $g$, which is used to explain the behaviour of the original model $f$ for a given instance $\mathbf{x}$. However, the effect of DSEG aligns with that of other segmentation methods like SLIC, as the surrogate model primarily leverages the resulting segments without incorporating additional elements from the segmentation foundation models underlying DSEG. This argument is further substantiated in the discussion of our experimental results.

## 4.2 HIERARCHICAL SEGMENTATION

The segmentation capabilities of foundation models like SAM, influenced by its design and hyperparameters, allow fine and coarse segmentation of an image (Kirillov et al., 2023). These models have the ability to segment a human-recognized concept at various levels, from the entirety of a car to its components, such as doors or windshields. This multitude of segments enables the composition of a concept into its sub-concepts, creating a hierarchical segmentation. We enhance the LIME framework by introducing hierarchical segmentation, allowing users to specify the granularity of the segment for more personalized explanations. The architecture allows the surrogate model to learn about features driven by human-recognizable concepts iteratively. DSEG starts by calculating the importance scores of the coarse segments in the first stage. The segments identified as highly important are subsequently refined into their finer components, followed by another importance score calculation. Next, we detail the steps involved in DSEG (as illustrated in Figure 2) to explain an image within the LIME framework, and Figure 3 shows the outputs of its intermediate steps. Additionally, the pseudocode for the proposed framework is presented in appendix A for clarity and reference.

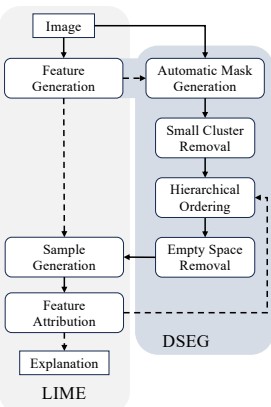

Figure 2: **Pipeline of DSEG in LIME.** Illustrating the LIME pipeline for image analysis with DSEG's specific steps - dashed lines represent the choice between applying DSEG or not, and is part of the feature generation process.

**Automatic mask generation.** Masks, also called segments or superpixels, represent distinct regions of an image. In the following, we denote the segmentation foundation model, such as SAM, by $\zeta$. Depending on the foundation model employed, $\zeta$ can be prompted using various methods, including points, area markings, text inputs, or automatically segmenting all visible elements in an image. For the main experiments of DSEG, we utilize the last prompt, automated mask generation, since we want to segment the whole image for feature generation without human intervention. We express the process as follows:

$$M_{\text{auto}} = \zeta(\mathbf{x}, G_{\text{prompt}}), \text{ with } \mathcal{S}^{\mathcal{D}} = M_{\text{auto}}, \tag{3}$$

where $\mathbf{x}$ denotes the input image, and $G_{\text{prompt}}$ specifies a general prompt configuration designed to enable automated segmentation. The output, $M_{\text{auto}}$, represents the automatically generated mask, as shown in Figure 3 (2). For this work, we used SAM with a grid overlay, parameterized by the number of points per side, to facilitate the automated segmentation process.

**Small cluster removal.** The underlying foundation model generates segments of varying sizes. We define a threshold $\theta$ such that segments with pixel-size below $\theta$ are excluded:

$$\mathcal{S}' = \{s_i \in \mathcal{S}^{\mathcal{D}} \mid \text{size}(s_i) \geq \theta\}. \tag{4}$$

In this study, we set $\theta = 500$ to reduce the feature set. The remaining superpixels in $\mathcal{S}'$ are considered for feature attribution. We incorporate this feature into DSEG to enable user-driven segment exclusion during post-processing, giving users control over the granularity within the segmentation hierarchy. This ensures that users can tailor the segmentation to their specific needs, thereby enhancing the method's flexibility and adaptability.

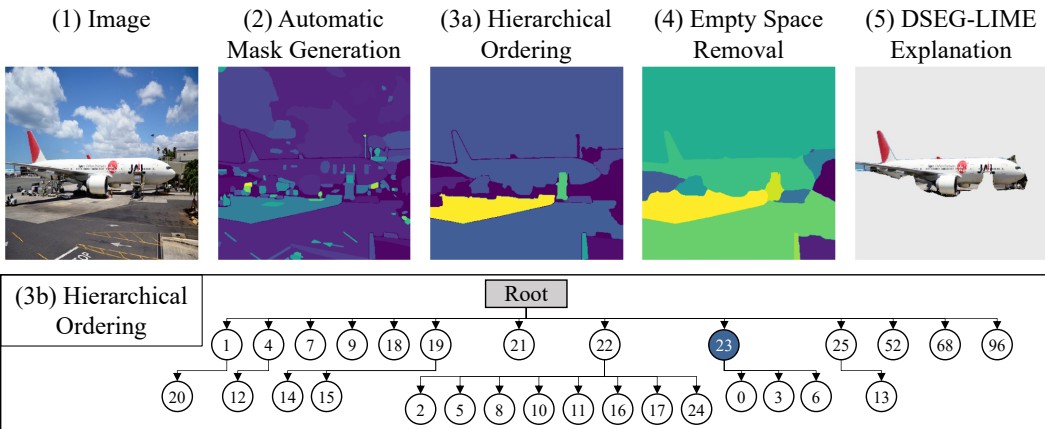

Figure 3: **Visualized DSEG pipeline.** Image (1) serves as the initial input, leading to its automatic segmentation depicted in (2). The hierarchical tree generated from this segmentation is illustrated in (3b), and (3a) showcasing the mask composed of first-order nodes. Image (4) displays the finalized mask created after eliminating empty spaces, which is fed back into the sample generation of LIME. Image (5) represents the resultant explanation within the DSEG-LIME framework. The image shows an instance from the COCO dataset (Lin et al., 2014), classified as an 'Airliner' ($p = 0.86$) by EfficientNetB4. Node 23 (blue node) indicates the segment that represents the superpixel of the airliner.

**Hierarchical ordering.** To handle overlapping segments, we impose a tree hierarchical structure $\mathcal{T} = (\mathcal{V}, \mathcal{E})$. In this structure, the overlap signifies that the foundation model has detected a sub-segment within a larger segment, representing the relationship between fine and coarse segments. The final output of DSEG, utilized for feature calculation, excludes overlapping segments. The nodes $v \in \mathcal{V}$ denote segments in $\mathcal{S}'$, and the edges $(u, v) \in \mathcal{E}$ encode the hierarchical relationship between segments. This hierarchical ordering process $H(\mathcal{S}')$ is a composition of the relative overlap of the segments, defined as:

$$H(\mathcal{S}') = \text{BuildHierarchy}(\mathcal{S}', \text{OverlapMetric}), \tag{5}$$

where OverlapMetric quantifies the extent of overlap between two segments $s_1, s_2 \in \mathcal{S}'$ defined by

$$\text{OverlapMetric}(s_1, s_2) = \frac{|s_1 \cap s_2|}{|s_2|}. \tag{6}$$

The hierarchy prioritizes parent segments (e.g., person) over child segments (e.g., clothing), as depicted (3a) in Figure 3. Each node represents one superpixel with its unique identifier. The depth $d$ of the hierarchy determines the granularity of the explanation, as defined by the user. A new set $\mathcal{S}'_d$, with $d = 1$, includes all nodes below the root. For $d > 1$, DSEG does not start from the beginning. Instead, it uses the segmentation hierarchy and segments $\mathcal{S}'$ from the first iteration. It then adds the nodes of the children of the top $k$ (a user-defined hyperparameter) most significant parent nodes in $\mathcal{S}'_d$ at depth $d - 1$, identified during the feature attribution phase. We visualize this selection in the tree shown in Figure 3 (3b), where all nodes with depth one, including the children of node 23, are considered in the second iteration. For the scope of this paper, we concentrate on the first-order hierarchy ($d = 1$) but provide additional explanations with $d = 2$ in the Appendix B.3.

**Empty space removal.** In hierarchical segmentation, some regions occasionally remain unsegmented. We refer to these areas as $R_{\text{unseg}}$. To address this, we employ the nearest neighbor algorithm, which assigns each unsegmented region in $R_{\text{unseg}}$ to the closest segment within the set $\mathcal{S}'_d$:

$$\mathcal{S}_d = \text{NearestNeighbor}(R_{\text{unseg}}, \mathcal{S}'_d). \tag{7}$$

Although this modifies the distinctiveness of concepts, it enhances DSEG-LIME's explanatory power. DSEG then utilizes the features $s_0, \ldots, s_d \in \mathcal{S}_d$ for feature attribution within LIME. Figure 3 (4) shows the corresponding mask along with the explanation of $d = 1$ in step (5) for the 'airliner' class. An ablation study of these steps is in Appendix C.1 and in Appendix C.5 we show exemplary feature attribution maps.

## 5 EVALUATION

In the following section, we will outline our experimental setup (Section 5.1) and introduce the XAI evaluation framework designed to assess DSEG-LIME both quantitatively (Section 5.2) and qualitatively (Section 5.3), compared to other LIME methodologies utilizing various segmentation algorithms. Subsequently, we discuss the limitations of DSEG (Section 5.4).

### 5.1 EXPERIMENTAL SETUP

**Segmentation algorithms.** Our experiment encompasses, along with SAM (vit_h), four conventional segmentation techniques: *Simple Linear Iterative Clustering* (SLIC) (Achanta et al., 2012), *Quickshift* (QS) (Hoyer et al., 2019), *Felzenszwalb* (FS) (Felzenszwalb & Huttenlocher, 2004) and *Watershed* (WS) (Neubert & Protzel, 2014). We carefully calibrate the hyperparameters of these techniques to produce segment counts similar to those generated by SAM. This calibration ensures that no technique is unfairly advantaged due to a specific segment count – for instance, scenarios where fewer but larger segments might yield better explanations than many smaller ones. In the Appendix, we demonstrate the universal property of integrating other segmentation methods within DSEG by presenting additional experiments with DETR (Carion et al., 2020) and SAM 2 (Ravi et al., 2024) in Appendix B.5, Appendix B.6.

**Models to explain.** The models investigated in this paper rely on pre-trained models, as our primary emphasis is on explainability. We chose EfficientNetB4 and EfficientNetB3 (Tan & Le, 2019) as the ones treated in this paper, where we explain EfficientNetB4 and use EfficientNetB3 for a contrastivity check (Nauta et al., 2023) (Section 5.2.1). To verify that our approach works on arbitrary pre-trained models, we also evaluated it using ResNet-101 (He et al., 2015; maintainers & contributors, 2016) (Appendix B.1) and VisionTransformer (ViT-384) (Dosovitskiy et al., 2020) (Appendix B.2). Furthermore, we demonstrate the applicability of our approach on a zero-shot learning example of CLIP (Radford et al., 2021) using a new dataset with other classes (Appendix B.7).

**Dataset.** We use images from the ImageNet classes (Deng et al., 2009), on which the covered models were trained (Tan & Le, 2019; He et al., 2015; Dosovitskiy et al., 2020). Our final dataset consists of 50 carefully selected instances (Appendix D.1), specifically chosen to comprehensively evaluate the techniques quantitatively. However, we want to emphasize that the selection of images is not biased toward any model. We also test the approach for another dataset in Appendix B.7. Additionally, our code (including documentation) is available in the supplementary material, allowing us to verify our claims. We will also make the code publicly available upon acceptance.

**Hyperparameters and hardware setup.** The experiments were conducted on an Nvidia RTX A6000 GPU. We compare standard LIME, SLIME (Zhou et al., 2021), GLIME (Tan et al., 2024), and BayLIME (Zhao et al., 2020), all integrated with DSEG, using 256 samples per instance, a batch size of ten and mean superpixel value for perturbation. For each explanation, up to three features are selected based on their significance, identified by values that exceed the average by more than 1.5 times the standard deviation. In BayLIME, we use the 'non-info-prior' setting. For SAM, we configure it to use 32 points per side, and conventional segmentation techniques are adjusted to achieve a similar segment count, as previously mentioned. In SLIC, we modify the number of segments and compactness; in Quickshift, the kernel size and maximum distance; in Felzenszwalb, the scale of the minimum size parameter; and in Watershed, the number of markers and compactness. Other hyperparameters remain at default settings to ensure a balanced evaluation across methods.

### 5.2 QUANTITATIVE EVALUATION

We adapt the framework by Nauta et al. (2023) to quantitatively assess XAI outcomes in this study, covering three domains: *content*, *presentation*, and *user experience*. In the content domain, we evaluate *correctness*, *output completeness*, *consistency*, and *contrastivity*. Presentation domain metrics like *compactness* and *confidence* are assessed under content for simplicity. We will briefly describe each metric individually to interpret the results correctly. The user domain, detailed in Section 5.3, includes a user study that compares our approach with other segmentation techniques in LIME. We use quantitative and qualitative assessments to avoid over-emphasizing technical precision or intuitive clarity (Molnar et al., 2022).

### 5.2.1 QUANTITATIVE METRICS DEFINITION

**Correctness** involves two randomization checks. The *model randomization parameter check* (Random Model) (Adebayo et al., 2020) tests if changing the random model parameters leads to different explanations. The *explanation randomization check* (Random Expl.) (Luo et al., 2020) examines if random output variations in the predictive model yield various explanations. For both metrics, in Table 1 we count the instances where explanations result in different predictions when reintroduced into the model under analysis. The domain also utilizes two deletion techniques: *single deletion* (Albini et al., 2020) and *incremental deletion* (Hoyer et al., 2019; Goyal et al., 2019). *Single deletion* serves as an alternative metric to assess the completeness of the explanation, replacing less relevant superpixels with a specific background to evaluate their impact on the model predictions (Ramamurthy et al., 2020). After these adjustments, we note instances where the model maintains the correct image classification. *Incremental deletion* (Incr. Deletion) entails progressively eliminating features from most to least significant based on their explanatory importance. We observe the model's output variations, quantifying the impact by measuring the area under the curve (AUC) of the model's confidence, as parts of the explanation are excluded. This continues until a classification change is observed (not ground truth class), and the mean AUC score for this metric is documented in Table 2.

**Output completeness** measures whether an explanation covers the crucial area for accurate classification. It includes a *preservation check* (Preservation) (Goyal et al., 2019) to assess whether the explanation alone upholds the original decision, and a *deletion check* (Deletion) (Zhang et al., 2023) to evaluate the effect of excluding the explanation on the prediction outcome (Ramamurthy et al., 2020). This approach assesses both the completeness of the explanation and its impact on the classification. The results are checked to ensure that the consistency of the classification is maintained. *Compactness* is also considered, highlighting that the explanation should be concise and cover all the areas necessary for prediction (Chang et al., 2019), reported by the mean value.

**Consistency** assesses explanation robustness to minor input alterations, like Gaussian noise addition, by comparing pre-and post-perturbation explanations for *stability against slight changes* (Noise Stability) (Zhang et al., 2021; Bhatt et al., 2021), using both preservation and deletion checks. For consistency of the feature importance score, we generate explanations for the same instance eight times (Rep. Stability), calculate the standard deviation $\sigma_i$ for each coefficient $i$, and then average all $\sigma_i$ values. This yields $\bar{\sigma}$, the average standard deviation of coefficients, and is reported as the mean score.

**Contrastivity** integrates several previously discussed metrics, aiming for *target-discriminative* explanations. This means that an explanation $e_{\mathbf{x}}$ for an instance $\mathbf{x}$ from a primary model $f_1$ (EfficientNetB4) should allow a secondary model $f_2$ (EfficientNetB3) to mimic the output of $f_1$ as $f_1(\mathbf{x}) \approx f_2(e_{\mathbf{x}})$ (Schwab & Karlen, 2019). The approach checks the explanation's utility and transferability across models, using EfficientNetB3 for preservation and deletion tests to assess consistency.

### 5.2.2 QUANTITATIVE EVALUATION RESULTS

Table 1 presents the outcomes of all metrics associated with class-discriminative outputs. The numbers in bold signify the top results, with an optimal score of 20. We compare LIME (L) (Ribeiro et al., 2016) with the LIME techniques discussed in Section 2, SLIME (S) (Zhou et al., 2021), GLIME (G) (Tan et al., 2024), and BayLIME (B) (Zhao et al., 2020) in combination with DSEG and the segmentation techniques from Section 5.1. The randomization checks in the correctness category confirm that the segmentation algorithm bias does not inherently affect any model. This is supported by the observation that most methods correctly misclassify when noise is introduced or the model's weights or predictions are shuffled. In contrast, DSEG excels in other metrics, surpassing alternative methods regardless of the LIME technique applied. In the output completeness domain, DSEG's explanations more effectively capture the critical areas necessary for the model to accurately classify an instance, whether by isolating or excluding the explanation. This efficacy is supported by the single deletion metric, akin to the preservation check but with a perturbed background. Moreover, noise does not compromise the consistency of DSEG's explanations. The contrastivity metric demonstrates DSEG's effectiveness in creating explanations that allow another AI model to produce similar outputs in over half of the cases and outperform alternative segmen-

tation approaches. Overall, the influence of the LIME feature attribution calculation does not vary much, because we only have an average of 15.56 segments in the covered dataset for the evaluation.

Table 1: **Quantitative summary - classes.** The table presents four quantitative areas and their metrics, comparing five segmentation techniques applied to EfficientNetB4: DSEG with SAM and comparative methods SLIC, Quickshift (QS), Felzenszwalb's (FS), and Watershed (WS). We test each with four LIME framework variations: LIME (L), SLIME (S), GLIME (G), and BayLIME (B). The experimental setup and metrics are detailed in Section 5.1 and Section 5.2.1. The table includes class-based metrics, with a maximum score of 50 for each; higher scores indicate better performance, and the highest scores for each metric are highlighted in bold.

| Domain | Metric | DSEG | | | | SLIC | | | | QS | | | | FS | | | | WS | | | |
|---|---|---|---|---|---|---|---|---|---|---|---|---|---|---|---|---|---|---|---|---|---|
| | | L | S | G | B | L | S | G | B | L | S | G | B | L | S | G | B | L | S | G | B |
| Correctness | Random Model ↑ | **38** | **38** | **38** | **38** | 30 | 30 | 30 | 30 | 35 | 35 | 34 | 34 | 36 | 36 | 36 | 36 | 33 | 33 | 33 | 33 |
| | Random Expl. ↑ | 40 | 41 | **46** | 44 | 38 | 45 | 39 | 38 | 39 | 34 | 42 | 38 | 42 | 39 | 40 | 38 | 36 | 39 | 39 | 36 |
| | Single Deletion ↑ | **28** | **29** | **29** | **29** | 18 | 17 | 21 | 21 | 13 | 13 | 11 | 11 | 18 | 17 | 18 | 19 | 13 | 13 | 14 | 13 |
| Output Completeness | Preservation ↑ | 36 | 38 | **41** | 40 | 37 | 35 | 35 | 35 | 33 | 32 | 32 | 33 | 37 | 38 | 39 | 38 | 38 | 36 | 37 | 37 |
| | Deletion ↑ | 31 | **33** | **33** | **33** | 21 | 21 | 21 | 21 | 17 | 17 | 17 | 17 | 21 | 20 | 21 | 23 | 22 | 22 | 21 | 22 |
| Consistency | Noise Stability ↑ | 36 | 36 | 36 | 36 | 35 | 36 | 36 | 36 | 28 | 28 | 29 | 27 | 38 | 36 | **39** | 37 | **39** | 38 | 38 | 37 |
| Contrastivity | Preservation ↑ | **31** | 29 | 30 | **31** | 28 | 28 | 27 | 28 | 19 | 20 | 18 | 19 | 27 | 27 | 28 | 28 | 26 | 27 | 27 | 27 |
| | Deletion ↑ | **33** | 32 | **33** | 32 | 23 | 24 | 24 | 24 | 22 | 22 | 22 | 22 | 23 | 22 | 24 | 23 | 22 | 22 | 22 | 22 |

Table 2 further illustrates DSEG's effectiveness in identifying key regions for model output, especially in scenarios of incremental deletion where SLIME outperforms. Bolded values represent the lowest numbers, signifying the best performance. Although compactness metrics show nearly uniform segment sizes across the techniques, Watershed's and Quickshift's smaller segments do not translate to better performance in other areas. Repeated experimentation suggests that stability is less influenced by the LIME variant and more by the segmentation approach, with SLIC and DSEG outperforming others. Further experiments show that DSEG outperforms SLIC regarding stability as the number of features increases. This advantage arises from the tendency of data-driven approaches to represent known objects uniformly as a single superpixel (Appendix C.3). Thus, if a superpixel accurately reflects the instance that the model in question predicts, it can be accurately and effortlessly matched with one or a few superpixels - such accurate matching leads to a more precise and more reliable explanation. Conventional segmentation algorithms often divide the same area into multiple superpixels, creating unclear boundaries and confusing differentiation between objects. The segmentation phase is the main differentiator regarding computation time; DSEG has longer processing times than the others (except Quickshift).

Table 2: **Quantitative summary - numbers.** The table summarizes metrics from Section 5.2.1, focusing on those quantified by rational numbers like incremental deletion, compactness, representational stability, and average computation time across the examples, detailed in Section 5.1. In contrast to Table 1, lower values indicate better performance and the lowest values are printed in bold.

| Metric | DSEG | | | | SLIC | | | | QS | | | | FS | | | | WS | | | |
|---|---|---|---|---|---|---|---|---|---|---|---|---|---|---|---|---|---|---|---|---|
| | L | S | G | B | L | S | G | B | L | S | G | B | L | S | G | B | L | S | G | B |
| Incr. Deletion ↓ | 1.25 | **0.38** | 0.40 | 0.37 | 0.68 | 0.70 | 0.75 | 0.69 | 1.46 | 1.44 | 1.40 | 1.38 | 1.45 | 1.42 | 1.45 | 1.41 | 0.76 | 0.74 | 0.76 | 0.76 |
| Compactness ↓ | 0.14 | 0.14 | 0.14 | 0.14 | 0.15 | 0.14 | 0.15 | 0.15 | 0.13 | **0.12** | 0.13 | 0.13 | 0.13 | 0.13 | 0.13 | 0.13 | **0.12** | 0.13 | **0.12** | 0.13 |
| Rep. Stability ↓ | **.010** | **.010** | .011 | **.010** | **.010** | **.010** | .011 | **.010** | .011 | .011 | .012 | **.010** | **.010** | .011 | .012 | .011 | .011 | .011 | .012 | .011 |
| Time ↓ | 32.4 | 29.8 | 36.7 | 32.0 | 22.9 | 24.5 | 27.6 | 25.6 | 45.1 | 49.6 | 50.1 | 46.4 | 19.9 | 20.2 | 22.7 | 22.1 | 16.9 | **16.1** | 17.5 | 17.7 |

## 5.3 QUALITATIVE EVALUATION

**User study.** Following the methodology by Chromik & Schuessler (2020), we conducted a user study (approved by the institute's ethics council) to assess the interpretability of the explanations. This study involved 87 participants recruited via Amazon Mechanical Turk (MTurk) and included 20 randomly of the 50 images in our dataset (Appendix D.1). These images were accompanied by explanations using DSEG and other segmentation techniques within the LIME framework (Section 5.1).

Table 3: **User study results.** This table summarizes each segmentation approach's average scores and top-rated counts of the user study results.

| Metric | DSEG | SLIC | QS | FS | WS |
|---|---|---|---|---|---|
| Avg. Score ↑ | **4.16** | 3.01 | 1.99 | 3.25 | 2.59 |
| Best Rated ↑ | **1042** | 150 | 90 | 253 | 205 |

Participants rated the explanations on a scale from 1 (least effective) to 5 (most effective) based on their intuitive understanding and the predicted class. Table 3 summarizes the average scores, the cumulative number of top-rated explanations per instance, and the statistical significance of user study results for each segmentation approach. DSEG is most frequently rated as the best and consistently ranks high even when it is not the leading explanation. Paired t-tests indicate that DSEG is statistically significantly superior (additional results in Appendix D.2).

### 5.4 LIMITATIONS AND FUTURE WORK

DSEG-LIME performs the feature generation directly on images before inputting them into the model for explanation. For models like ResNet with smaller input sizes (He et al., 2015), the quantitative advantages of DSEG are less evident (Appendix B.1). Experiments have shown that better results can be achieved with a lower stability score threshold of SAM. Furthermore, substituting superpixels with a specific value in preservation and deletion evaluations can introduce an inductive bias (Garreau & Mardaoui, 2021). To reduce this bias, using a generative model to synthesize replacement areas could offer a more neutral alteration. Additionally, future work should thoroughly evaluate feature attribution maps to ensure that methods assign significant attributions to the correct regions, like in appendix C.5. This comprehensive assessment is essential for verifying the interpretability and reliability of such methods. Lastly, our approach, like any other LIME-based method (Ribeiro et al., 2016; Zhou et al., 2021; Zhao et al., 2020; Tan et al., 2024), does not assume a perfect match between the explanation domains and the model's actual domains since it simplifies the model by a local surrogate. Nonetheless, our quantitative analysis confirms that the approximations closely reflect the model's behaviour. Future work could focus on integrating the foundation model directly into the system through a model-intrinsic approach, similar to (Sun et al., 2023).

**No free lunch.** Although DSEG provides promising results in many domains, it is not always universally applicable. When domain-specific knowledge is crucial to identify meaningful features or the feature generation task is inherently complex, DSEG might not perform as effectively as traditional segmentation methods within LIME (Khani et al., 2024) (Appendix C.4). However, future exploration could involve testing alternative segmentation techniques, such as integrating HSSN (Wang et al., 2023) or HIPPIE (Li et al., 2022) instead of SAM (or DETR) to overcome this limitation.

## 6 CONCLUSION

In this study, we introduced DSEG-LIME, an extension to the LIME framework, incorporating a data-driven foundation model (SAM) for feature generation. This approach ensures that the generated features more accurately reflect human-recognizable concepts, enhancing the interpretability of explanations. Furthermore, we refined the process of feature attribution within LIME through an iterative method, establishing a segmentation hierarchy that contains the relationships between components and their subcomponents. In Appendix C.2, we show that our idea also helps explain a model's wrong classifications. Through a comprehensive two-part evaluation, split into quantitative and qualitative analysis, DSEG emerged as the superior method, outperforming other LIME-based approaches in most evaluated metrics. The adoption of foundational models marks a significant step towards enhancing the post-hoc and model-agnostic interpretability of deep learning models.

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

# Appendix

## Table of Contents

## A DSEG-LIME ALGORITHM

---

**Algorithm 1** DSEG-LIME framework pseudocode

---

**Require:** $f$ (black-box model), $\zeta$ (segmentation function), $x$ (input instance), $g$ (interpretable model), $d$ (maximum depth), $hp$ (segmentation hyperparameters), $\theta$ (minimum segment size), $k$ (top segments to select)

**Ensure:** $g$ approximates $f$ locally around $x$

  **1. Initial segmentation:**
  $\mathcal{S} \leftarrow \zeta(x, hp)$                                ▷ Segment the input instance
  **2. Small cluster removal:**
  $\mathcal{S}' \leftarrow \{s_i \in \mathcal{S} \mid \text{size}(s_i) \geq \theta\}$                      ▷ Remove small clusters
  **3. Hierarchical ordering:**
  $\mathcal{H} \leftarrow \text{BuildHierarchy}(\mathcal{S}')$                ▷ Build hierarchical segmentation
  **for** $l \leftarrow 1$ to $d$ **do**
    **4. Empty space removal:**
    $\mathcal{H}[l] \leftarrow \text{NearestNeighbor}(\mathcal{H}[l])$          ▷ Fill unsegmented space
    **if** $l = 1$ **then**
      $\mathcal{S}_l \leftarrow \mathcal{H}[l]$                         ▷ Segments at depth 1
    **else**
      $\mathcal{S}_l \leftarrow \{s_i \in \mathcal{H}[l] \mid \text{parent}(s_i) \in top\_ids\}$   ▷ Select child segments of top parents
    **end if**
    $Z \leftarrow \text{Perturb}(x, \mathcal{S}_l)$                ▷ Create neighborhood perturbations
    $w \leftarrow \text{Proximity}(Z, x)$        ▷ Compute sample weights based on proximity
    $\text{preds} \leftarrow f(z)$ for all $z \in Z$            ▷ Get predictions from $f$
    $g \leftarrow \text{InitializeModel}(g)$       ▷ Initialize a new interpretable model for depth $l$
    $g \leftarrow \text{Fit}(g, Z, \text{preds}, w)$             ▷ Train interpretable model
    $top\_ids \leftarrow \{\text{id}(s_i) \mid s_i \in \mathcal{S}_l, s_i \text{ is among top } k \text{ features in } g\}$   ▷ Update top segment IDs
  **end for**
  **return** $g$                            ▷ Return the local surrogate model

---

We present the pseudocode of our DSEG-LIME framework in Algorithm 1. To construct the hierarchical segmentation within our framework, we start by calculating the overlaps between all segments in $\mathcal{S}$. We build a hierarchical graph using this overlap information through the following process.

First, we identify the top-level segments, which do not occur as subparts of any other segments. These segments serve as the highest-level nodes in the hierarchical graph. Starting from these top-level segments, we apply a top-down approach to identify child segments recursively. For each parent segment, we check for segments that are contained within it; these fully contained segments are designated as child nodes of the parent in the graph.

This recursive process continues for each subsequent level, ensuring that every parent node encompasses its child nodes. The hierarchical graph thus formed represents the structural relationships between segments, where parent-child relationships indicate that child segments are complete parts of their respective parent segments. By constructing the hierarchy in this manner, we capture the nested structure of segments, which supports multi-level interpretability within the DSEG-LIME framework.

# B    Supplementary model evaluations

## B.1    ResNet

In Table 4, we detail the quantitative results for the ResNet-101 model, comparing our evaluation with the criteria used for EfficientNetB4 under consistent hyperparameter settings. The review extends to a comparative analysis with EfficientNetB3, focusing on performance under contrastive conditions. The results confirm the EfficientNet results and show that the LIME techniques behave unpredictably in the presence of model noise or prediction shuffle despite different segmentation strategies. This indicates an inherent randomness in the model explanations. The single deletion metric showed that all XAI approaches performed below EfficientNetB4, with DSEG performing slightly better than its counterparts. However, DSEG performed best on other metrics, especially when combined with the SLIME framework, where it showed superior resilience to noise, distinguishing it from alternative methods.

Table 4: **Quantitative summary - classes ResNet-101.** The table presents the metrics consistently with those discussed for EfficientNet.

| Domain | Metric | DSEG | | | | SLIC | | | | QS | | | | FS | | | | WS | | | |
|---|---|---|---|---|---|---|---|---|---|---|---|---|---|---|---|---|---|---|---|---|---|
| | | L | S | G | B | L | S | G | B | L | S | G | B | L | S | G | B | L | S | G | B |
| Correctness | Random Model ↑ | 45 | 45 | 43 | 45 | 42 | 42 | 41 | 44 | 44 | 44 | 45 | 44 | 40 | 47 | 45 | 44 | 46 | 44 | 44 | **49** |
| | Random Expl. ↑ | 49 | 48 | 49 | 47 | 45 | 47 | 46 | 46 | 45 | 46 | **50** | 47 | 46 | 48 | 48 | 43 | 44 | 45 | 47 | 46 |
| | Single Deletion ↑ | 10 | 10 | 6 | **12** | 7 | 9 | 6 | 7 | 4 | 5 | 5 | 5 | 7 | 7 | 5 | 7 | 8 | 6 | 8 | 7 |
| Output Completeness | Preservation ↑ | 20 | **22** | 18 | 20 | 21 | 21 | 16 | 17 | 12 | 12 | 13 | 10 | 15 | 14 | 17 | 14 | 19 | 17 | 18 | 13 |
| | Deletion ↑ | 27 | **30** | 27 | 31 | 18 | 16 | 15 | 16 | 20 | 17 | 17 | 19 | 18 | 20 | 21 | 18 | 18 | 19 | 17 | 15 |
| Consistency | Noise Stability ↑ | **20** | **20** | 15 | 14 | 17 | 13 | 15 | 18 | 13 | 14 | 15 | 9 | 12 | 16 | 14 | 14 | 18 | 17 | 19 | 15 |
| Contrastivity | Preservation ↑ | 17 | **22** | 18 | 18 | 13 | 14 | 13 | 14 | 19 | 17 | 10 | 19 | 19 | 22 | 21 | 20 | 21 | 18 | 20 | 15 |
| | Deletion ↑ | 25 | 28 | 28 | **30** | 23 | 23 | 24 | 22 | 24 | 21 | 21 | 23 | 22 | 22 | 21 | 22 | 18 | 19 | 19 | 18 |

Table 5 presents further findings of ResNet. SLIME with DSEG yields the lowest AUC for incremental deletion, whereas Quickshift and Felzenszwalb show the highest. WS produces the smallest superpixels for compactness, contrasting with DSEG's larger ones. The stability analysis shows that all segmentations are almost at the same level, with SLIC being the best and GLIME the best-performing overall. Echoing EfficientNet's review, segmentation defines runtime, with DSEG being the most time-consuming. The runtime disparities between the ResNet and EfficientNet models are negligible.

Table 5: **Quantitative summary - numbers ResNet-101.** The table presents the metrics consistently with those discussed for EfficientNet.

| Metric | DSEG | | | | SLIC | | | | QS | | | | FS | | | | WS | | | |
|---|---|---|---|---|---|---|---|---|---|---|---|---|---|---|---|---|---|---|---|---|
| | L | S | G | B | L | S | G | B | L | S | G | B | L | S | G | B | L | S | G | B |
| Incr. Deletion ↓ | 0.54 | 0.28 | 0.27 | **0.26** | 0.55 | 0.54 | 0.56 | 0.55 | 0.90 | 0.84 | 0.92 | 0.86 | 0.85 | 0.81 | 0.89 | 0.88 | 0.50 | 0.50 | 0.49 | 0.49 |
| Compactness ↓ | 0.25 | 0.20 | 0.24 | 0.24 | 0.16 | 0.16 | 0.16 | 0.16 | 0.14 | 0.16 | 0.14 | 0.15 | 0.16 | 0.16 | 0.15 | 0.16 | **0.13** | **0.13** | **0.13** | 0.14 |
| Rep. Stability ↓ | .021 | .021 | .018 | .022 | .019 | .018 | .016 | .019 | .018 | .018 | **.015** | .018 | .018 | .018 | .016 | .018 | .017 | .018 | .016 | .018 |
| Time ↓ | 8.0 | 8.2 | 8.1 | 8.4 | 2.8 | 3.0 | 2.7 | **2.5** | 12.6 | 13.3 | 13.5 | 12.9 | 2.9 | 2.7 | 2.8 | 3.0 | 3.5 | 2.9 | 2.9 | 3.2 |

## B.2    VisionTransformer

Table 6 provides the quantitative results for the VisionTransformer (ViT-384) model, employing settings identical to those used for EfficientNet and ResNet, with ViT processing input sizes of (384x384). The class-specific results within this table align closely with the performances recorded for the other models, further underscoring the effectiveness of DSEG. This consistency in DSEG performance is also evident in the data presented in Table 7. However, the 'Noise Stability' metric shows poorer performance for both models than for EfficientNetB4, indicating that ViT and ResNet have greater difficulty when noise enters the input.

We performed all experiments for ResNet and ViT with the same hyperparameters defined for EfficientNetB4. We would like to explicitly point out that the quantitative results could be improved by

defining more appropriate hyperparameters for both DSEG and conventional segmentation methods, as no hyperparameter search was performed for a fair comparison.

Table 6: **Quantitative summary - classes ViT-384.** The table presents the metrics consistently with those discussed for EfficientNet.

| Domain | Metric | DSEG | | | | SLIC | | | | QS | | | | FS | | | | WS | | | |
|---|---|---|---|---|---|---|---|---|---|---|---|---|---|---|---|---|---|---|---|---|---|
| | | L | S | G | B | L | S | G | B | L | S | G | B | L | S | G | B | L | S | G | B |
| Correctness | Random Model ↑ | 44 | 44 | 44 | 44 | 44 | 44 | 43 | 44 | **46** | **46** | **46** | **46** | 44 | 44 | 44 | 43 | 42 | 42 | 42 | 42 |
| | Random Expl. ↑ | 46 | 48 | **50** | 49 | 46 | 46 | 46 | 46 | **50** | 48 | 49 | 49 | 48 | 49 | 46 | 48 | 45 | 45 | 47 | 44 |
| | Single Deletion ↑ | 19 | **20** | **20** | **20** | 14 | 12 | 13 | 12 | 11 | 10 | 10 | 12 | 11 | 12 | 13 | 13 | 13 | 12 | 14 | 14 |
| Output Completeness | Preservation ↑ | 15 | 15 | 15 | 15 | **17** | **17** | 16 | 16 | 11 | 9 | 11 | 12 | 14 | 14 | 13 | 13 | 16 | 15 | 16 | 16 |
| | Deletion ↑ | **39** | 36 | 36 | 37 | 34 | 33 | 34 | 33 | 30 | 29 | 31 | 31 | 32 | 31 | 32 | 31 | 31 | 31 | 32 | 32 |
| Consistency | Noise Stability ↑ | 16 | 12 | 12 | 12 | **20** | 18 | 18 | 16 | 9 | 11 | 11 | 9 | 11 | 10 | 11 | 11 | 14 | 16 | 17 | 16 |
| Contrastivity | Preservation ↑ | 28 | 27 | **30** | 27 | 22 | 22 | 23 | 22 | 23 | 21 | 25 | 21 | **30** | 29 | 28 | **30** | 24 | 23 | 23 | 24 |
| | Deletion ↑ | 32 | **33** | 32 | 32 | 25 | 25 | 25 | 26 | 27 | 27 | 27 | 27 | 24 | 25 | 24 | 24 | 25 | 26 | 26 | 26 |

Table 7: **Quantitative summary - numbers ViT-384.** The table presents the metrics consistently with those discussed for EfficientNet.

| Metric | DSEG | | | | SLIC | | | | QS | | | | FS | | | | WS | | | |
|---|---|---|---|---|---|---|---|---|---|---|---|---|---|---|---|---|---|---|---|---|
| | L | S | G | B | L | S | G | B | L | S | G | B | L | S | G | B | L | S | G | B |
| Incr. Deletion ↓ | 1.01 | 0.53 | 0.49 | **0.35** | 0.78 | 0.85 | 0.80 | 0.82 | 1.63 | 1.63 | 1.66 | 1.60 | 1.62 | 1.69 | 1.69 | 1.73 | 1.00 | 0.99 | 1.01 | 1.03 |
| Compactness ↓ | 0.19 | 0.18 | 0.18 | 0.18 | 0.16 | 0.15 | 0.15 | 0.15 | 0.12 | 0.12 | 0.12 | 0.12 | 0.14 | 0.14 | 0.14 | 0.14 | **0.11** | **0.11** | **0.11** | **0.11** |
| Rep. Stability ↓ | **.014** | **.014** | **.014** | **.014** | **.014** | **.014** | .015 | **.014** | .017 | .017 | .018 | .017 | .015 | .015 | .015 | .015 | .016 | .016 | .017 | .016 |
| Time ↓ | 8.2 | 7.4 | 8.1 | 7.6 | 2.4 | **2.2** | 2.5 | 2.5 | 13.6 | 13.1 | 13.7 | 13.0 | 4.9 | 4.7 | 5.6 | 4.5 | 2.7 | **2.2** | 2.5 | 2.6 |

## B.3 EFFICIENTNETB4 WITH DEPTH OF TWO

In Table 8 and Table 9, we present the quantitative comparison between DSEG-LIME ($d = 2$) using EfficientNetB4 and SLIC, as reported in the main paper. The hyperparameter settings were consistent across the evaluations, except for compactness. We established a minimum threshold of 0.05 for values to mitigate the impact of poor segmentation performance, which often resulted in too small segments. Additional segments were utilized to meet this criterion for scenarios with suboptimal segmentation. However, this compactness constraint was not applied to DSEG with depth two since its hierarchical approach naturally yields smaller and more detailed explanations, evident in Table 9. The hierarchical segmentation of $d = 2$ slightly impacts stability, yet the method continues to generate meaningful explanations, as indicated by other metrics. Although our method demonstrated robust performance, it required additional time because the feature attribution process was conducted twice.

Table 8: **Quantitative summary - classes depth two.** The table showcases metrics for EfficientNetB4, specifically at a finer concept granularity; the hierarchical segmentation tree has $d = 2$. Results reported pertain solely to integrating DSEG and SLIC within the scope of the LIME frameworks examined.

| Domain | Metric | DSEG | | | | SLIC | | | |
|---|---|---|---|---|---|---|---|---|---|
| | | L | S | G | B | L | S | G | B |
| Correctness | Random Model ↑ | **36** | **36** | **36** | 35 | 30 | 30 | 30 | 30 |
| | Random Expl. ↑ | 41 | 42 | 39 | 42 | 38 | **45** | 39 | 38 |
| | Single Deletion ↑ | 20 | **22** | 21 | 21 | 18 | 17 | 21 | 21 |
| Output Completeness | Preservation ↑ | 36 | 34 | 36 | 35 | **37** | 35 | 35 | 35 |
| | Deletion ↑ | 21 | **27** | 26 | **27** | 21 | 21 | 21 | 21 |
| Consistency | Noise Stability ↑ | 31 | 36 | **40** | 37 | 35 | 36 | 36 | 36 |
| Contrastivity | Preservation ↑ | **31** | 29 | **31** | 29 | 28 | 28 | 27 | 28 |
| | Deletion ↑ | 21 | 22 | 21 | 22 | 23 | **24** | **24** | **24** |

Table 9: **Quantitative summary - numbers depth two.** The table showcases the numeric values in the same manner as in Table 8 but for numeric values.

| Metric | DSEG | | | | SLIC | | | |
|---|---|---|---|---|---|---|---|---|
| | L | S | G | B | L | S | G | B |
| Incr. Deletion ↓ | 1.19 | 0.79 | **0.54** | 1.11 | 0.68 | 0.70 | 0.75 | 0.69 |
| Compactness ↓ | 0.16 | 0.18 | 0.15 | 0.18 | 0.15 | **0.14** | 0.15 | 0.15 |
| Rep. Stability ↓ | .012 | .012 | .013 | .012 | **.010** | **.010** | .011 | **.010** |
| Time ↓ | 47.6 | 52.5 | 53.4 | 52.8 | **22.9** | 24.5 | 27.6 | 25.6 |

**Exemplary explanations.** DSEG-LIME introduces a hierarchical feature generation approach, allowing users to specify segmentation granularity via tree depth. Figure 4 displays five examples from our evaluation, with the top images showing DSEG's explanations at a hierarchy depth of one and the bottom row at a depth of two. These explanations demonstrate that deeper hierarchies focus on smaller regions. However, the banana example illustrates a scenario where no further segmentation occurs if the concept, like a banana, lacks sub-components for feature generation, resulting in identical explanations at both depths.

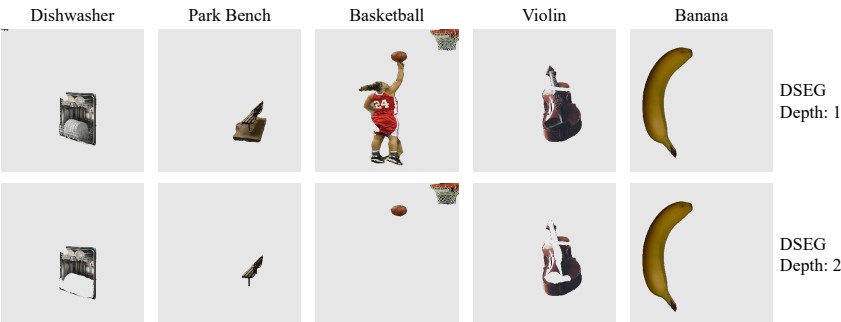

Figure 4: **DSEG depth two.** The figure displays exemplary images from the evaluation dataset, illustrating DSEG explanations at $d = 2$ of hierarchical segmentation. These images serve as complementary examples to the paper's discussion on the projectile, enhancing the illustration of the concept.

In Figure 5, another instance is explained with DSEG and $d = 2$, showing a black-and-white image of a projectile. Here, we see the corresponding explanation for each stage, starting with the first iteration with the corresponding segmentation map. In the second iteration, we see the segment representing the projectile split into its finer segments - the children nodes of the parent node - with the corresponding explanation below.

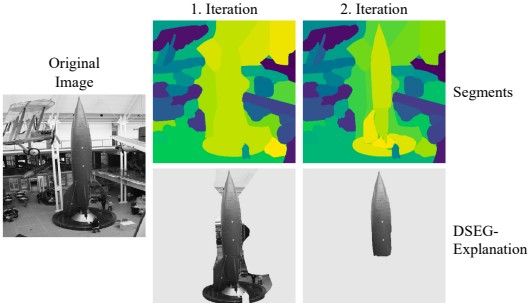

Figure 5: **2nd iteration of DSEG-LIME.** Visualizing DSEG's explanations of a projectile. It includes the first iteration's explanation along with its corresponding segmentation map. Additionally, similar details are provided for the second iteration procedure, highlighting the upper part of a projectile as an explanation.

**Case study.** We examine the case presented in Figure 3, where DSEG initially segments the image into various layers with overlapping features, establishing a segmentation hierarchy through composition. In the first iteration, LIME focuses solely on the segments just beneath the root node - the parent segments that cannot be merged into broader concepts. From this segmentation map, LIME determines the feature importance scores, identifying the airplane as the most crucial element in the image. In the subsequent iteration, illustrated in Figure 6, DSEG generates an additional segmentation map that further divides the airplane into finer components for detailed analysis. The explanation in this phase emphasizes the airplane's body, suggesting that this concept of the 'Airliner' is most significant.

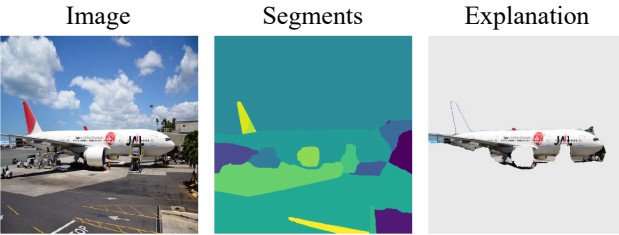

Figure 6: **Airliner explanation with depth two.** The same example as in Figure 3 but with segmentation hierarchy of two for the explanation. This example includes the children nodes of the most significant parent node in the segmentation map for feature importance calculation.

## B.4 DSEG COMPARED TO EAC

We conducted additional experiments with Explain any Concept (EAC) (Sun et al., 2023), performing the same quantitative experiments as for DSEG. We began our evaluation by noting that EAC, unlike DSEG-LIME, cannot be applied to arbitrary models, which is a significant drawback of their method and prevents comprehensive comparisons. Thus, we compared our approach against LIME and EAC, in explaining ResNet. The results are listed in Table 10 and Table 11.

Table 10: **Quantitative summary - classes EAC.** This table presents the metrics in line with the previous evaluations, focusing on ResNet performance for DSEG and other segmentation techniques in comparison to EAC.

| Domain | Metric | LIME-DSEG | LIME-SLIC | LIME-QS | LIME-FS | LIME-WS | EAC |
|---|---|---|---|---|---|---|---|
| | Random Model ↑ | 45 | 42 | 44 | 40 | **46** | 45 |
| Correctness | Random Expl. ↑ | **49** | 45 | 45 | 46 | 44 | 48 |
| | Single Deletion ↑ | **10** | 7 | 4 | 7 | 8 | 1 |
| Output | Preservation ↑ | 20 | 21 | 12 | 15 | 19 | **35** |
| Completeness | Deletion ↑ | 27 | 18 | 20 | 18 | 18 | **38** |
| Consistency | Noise Stability ↑ | 20 | 17 | 13 | 12 | 18 | **34** |
| Contrastivity | Preservation ↑ | 17 | 13 | 19 | 19 | 21 | **31** |
| | Deletion ↑ | 25 | 23 | 24 | 22 | 18 | **35** |

We observe that EAC quantitatively outperforms DSEG in certain cases. However, the results indicate that DSEG shows marked improvement as the number of samples increases, ultimately achieving comparable computation times. Moreover, it is expected that EAC performs better with ResNet, as it is specifically designed to leverage the model's internal representations. The main drawback of EAC, however, is its lack of general applicability, as it cannot be used across all model architectures.

Table 11: **Quantitative summary - numbers EAC.** This table presents the metrics in line with the previous evaluations, focusing on ResNet performance for DSEG and other segmentation techniques in comparison to EAC.

| Metric | LIME-DSEG | LIME-SLIC | LIME-QS | LIME-FS | LIME-WS | EAC |
|---|---|---|---|---|---|---|
| Incr. Deletion ↓ | 0.54 | 0.55 | 0.90 | 0.85 | 0.50 | **0.01** |
| Compactness ↓ | 0.25 | 0.16 | 0.14 | 0.16 | 0.13 | **0.11** |
| Rep. Stability ↓ | .021 | .019 | .018 | .018 | .017 | **.002** |
| Time ↓ | 8.0 | **2.8** | 12.6 | 2.9 | 3.5 | 326.7 |

## B.5 DETR WITHIN DSEG

In Table 12 and Table 13, we conducted the DETR experiments within LIME. Based on previous results, we evaluate its performance by comparing it to SLIC within LIME. Both experiments were configured with identical parameters, and DETR was implemented for basic panoptic segmentation.

Table 12: **Quantitative summary - classes DETR.** The table showcases metrics for EfficientNetB4, specifically at a finer concept granularity; the hierarchical segmentation tree has a depth of two. Results reported pertain solely to integrating DSEG and SLIC within the scope of the LIME frameworks examined.

| Domain | Metric | DSEG | | | | SLIC | | | |
|---|---|---|---|---|---|---|---|---|---|
| | | L | S | G | B | L | S | G | B |
| Correctness | Random Model ↑ | **32** | **32** | **32** | **32** | 30 | 30 | 30 | 30 |
| | Random Expl. ↑ | 29 | 37 | 38 | 40 | 38 | **45** | 39 | 38 |
| | Single Deletion ↑ | **36** | **36** | 35 | **36** | 18 | 17 | 21 | 21 |
| Output Completeness | Preservation ↑ | **43** | 42 | 42 | 42 | 37 | 35 | 35 | 35 |
| | Deletion ↑ | 34 | 34 | **35** | 34 | 21 | 21 | 21 | 21 |
| Consistency | Noise Stability ↑ | **40** | **40** | 39 | 39 | 35 | 36 | 36 | 36 |
| Contrastivity | Preservation ↑ | **39** | 37 | 36 | 36 | 28 | 28 | 27 | 28 |
| | Deletion ↑ | **35** | 34 | 33 | 32 | 23 | 24 | 24 | 24 |

Table 13: **Quantitative summary - numbers DETR.** The table showcases the numeric values in the same manner as in Table 12 but for numeric values.

| Metric | DSEG | | | | SLIC | | | |
|---|---|---|---|---|---|---|---|---|
| | L | S | G | B | L | S | G | B |
| Incr. Deletion ↓ | 0.64 | 0.34 | 0.37 | **0.25** | 0.68 | 0.70 | 0.75 | 0.69 |
| Compactness ↓ | 0.34 | 0.34 | 0.34 | 0.34 | 0.15 | **0.14** | 0.15 | 0.15 |
| Rep. Stability ↓ | .008 | .008 | .008 | **.007** | .010 | .010 | .011 | .010 |
| Time ↓ | 23.6 | **22.0** | 24.4 | 23.5 | 22.9 | 24.5 | 27.6 | 25.6 |

DETR demonstrates superior performance on the dataset compared to the LIME variants utilizing SLIC. Despite its efficacy, the segmentation quality of DETR was generally inferior to that of SAM, as evidenced by less compact explanations. This observation is further supported by the examples in Figure 7. The visualizations reveal that DETR often segments images in ways that do not align with typical human-recognizable concepts, highlighting a potential limitation in its practical utility for generating explanatory segments. Moreover, DETR does not support the construction of a segmentation hierarchy, lacking the ability to produce finer and coarser segments, which diminishes its flexibility compared to methods such as SAM.

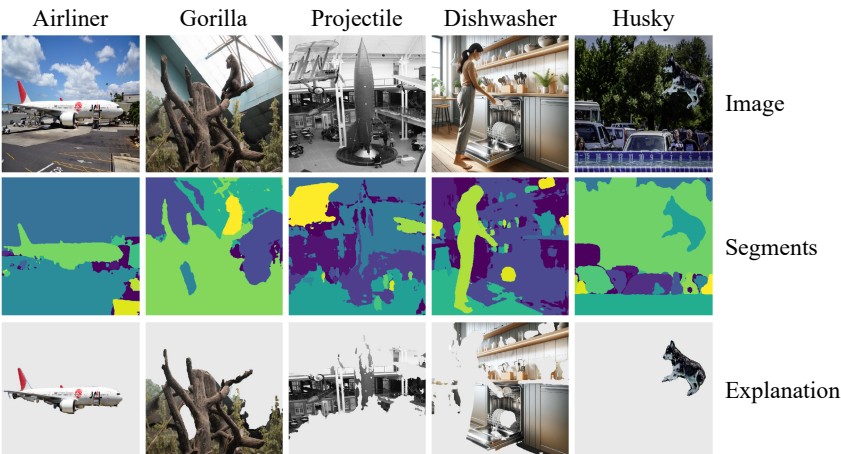

Figure 7: **DETR within DSEG.** The visualization displays five instances with classes from the ImageNet dataset. Each image includes the prediction by EfficientNetB4 as its headline, the segmentation map of DETR, and the corresponding explanation by DETR within LIME.

## B.6 DSEG-LIME WITH SAM 2

In the main paper, we conducted experiments using SAM 1. In this part, we integrate SAM 2 (Ravi et al., 2024) with the 'hiera_l' backbone into the DSEG framework, applying a 0.8 stability score threshold. The results for EfficientNetB4 are presented in Table 14 and Table 15.

Table 14: **Quantitative summary - classes SAM 2.** This table presents the metrics in line with those discussed for EfficientNet in the main paper, but displays only the results for SLIC for the sake of simplicity.

| Domain | Metric | DSEG | | | | SLIC | | | |
|---|---|---|---|---|---|---|---|---|---|
| | | L | S | G | B | L | S | G | B |
| Correctness | Random Model ↑ | **38** | **38** | **38** | **38** | 30 | 30 | 30 | 30 |
| | Random Expl. ↑ | 40 | 44 | 43 | 44 | 38 | **45** | 39 | 38 |
| | Single Deletion ↑ | 27 | 27 | **28** | **28** | 18 | 17 | 21 | 21 |
| Output Completeness | Preservation ↑ | **39** | 34 | 35 | 34 | 37 | 35 | 35 | 35 |
| | Deletion ↑ | **34** | 30 | 30 | 30 | 21 | 21 | 21 | 21 |
| Consistency | Noise Stability ↑ | **38** | **38** | **38** | 37 | 35 | 36 | 36 | 36 |
| Contrastivity | Preservation ↑ | **31** | 28 | 29 | 30 | 28 | 28 | 27 | 28 |
| | Deletion ↑ | **35** | 30 | 31 | 31 | 23 | 24 | 24 | 24 |

As both tables demonstrate, DSEG-LIME consistently outperforms other methods and surpasses DSEG with SAM 1 across most metrics, delivering superior results. It effectively segments images into more meaningful regions, particularly in cases where SAM 1 faced challenges, reinforcing the conclusions of the SAM 2 technical report.

However, since the experiments were conducted on different hardware, the computation times vary. Here, we report the time for the SLIC variant of LIME, but similar to previous experiments, the times for other LIME variants with SLIC are expected to be comparable to those of standard LIME. As a result, DSEG with SAM 2 is slightly slower due to the additional segmentation process.

Table 15: **Quantitative summary - numbers SAM 2.** This table presents the metrics in line with those discussed for EfficientNet in the main paper, but displays only the results for SLIC for the sake of simplicity.

| Metric | DSEG | | | | SLIC | | | |
|---|---|---|---|---|---|---|---|---|
| | L | S | G | B | L | S | G | B |
| Incr. Deletion ↓ | 0.98 | **0.36** | **0.36** | 0.39 | 0.68 | 0.70 | 0.75 | 0.69 |
| Compactness ↓ | 0.16 | 0.16 | 0.17 | 0.17 | 0.15 | **0.14** | 0.15 | 0.15 |
| Rep. Stability ↓ | .011 | **.010** | .011 | **.010** | **.010** | **.010** | .011 | **.010** |
| Time ↓ | 19.1 | 18.9 | 18.7 | 19.3 | **14.7** | - | - | - |

**Exemplary explanations.** Figure 8 presents explanations generated by DSEG using both SAM 1 and SAM 2, highlighting cases where the newer version of SAM enables DSEG to produce more meaningful and interpretable explanations. Each image includes explanations for the predicted class from EfficientNetB4. While SAM 2 shows improved segmentation in these examples, similar results can be obtained with SAM 1 by appropriately adjusting the hyperparameters for automatic mask generation.

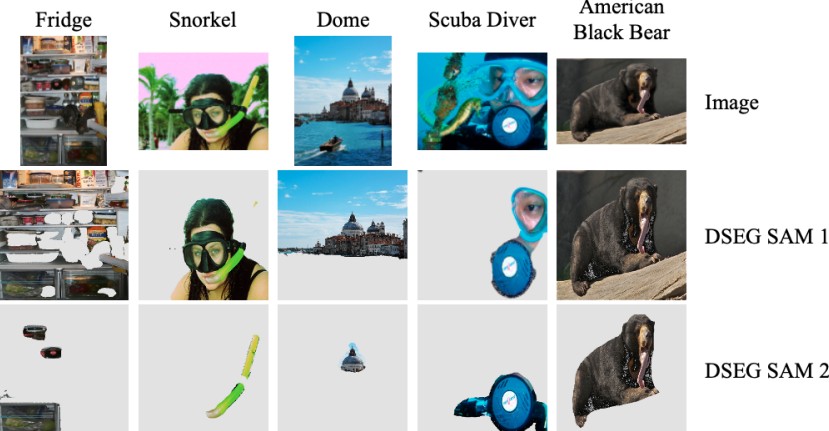

Figure 8: **Comparison DSEG with SAM 1 and SAM 2.** Exemplary images with explanations generated by SAM 1 and SAM 2 within DSEG, illustrating how the updated SAM improves segment utilization for DSEG.

### B.7 ZERO-SHOT CLASSIFICATION EXPLANATION

In this section, we demonstrate the versatility of DSEG-LIME by applying it to a different dataset and classification task. Specifically, we replicate the zero-shot classification approach described in (Prasse et al., 2023) using CLIP (Radford et al., 2021) for the animal super-category. Since DSEG-LIME maintains model-agnostic properties, it remains applicable to zero- and few-shot classification models without modification.

Figure 9 presents an illustrative example from the dataset, where the task is to classify an image into the animal category. The predicted and ground-truth class for the image is 'Land mammal'. As shown by DSEG-LIME's explanation, the model's decision is primarily influenced by the presence of a deer in the foreground and a mountain in the background, which contribute to the overall classification.

Original Image    DSEG Explanation

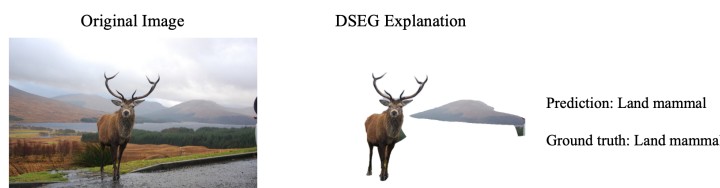

Prediction: Land mammal

Ground truth: Land mammal

Figure 9: **DSEG-LIME explanation for CLIP.** This figure illustrates an image processed by CLIP for zero-shot classification into animal categories. The model correctly predicted the class as "Land mammal." DSEG-LIME highlights the two most important features influencing the classification, with the presence of the deer being the most significant.

## C    FURTHER EXPERIMENTS WITH DSEG-LIME

### C.1    ABLATION STUDY

For the ablation study, we examine how the number of segments evolves across different stages of the segmentation process as we vary the threshold for removing segments smaller than the hyper-parameter $\theta$ (with values [100, 300, 500, 1000, 2000]). Additionally, we assess the behavior of empty spaces within the segmented regions across all images in the dataset. The analysis focuses on three key points: the number of segments immediately after the initial automated segmentation, after hierarchical sorting, and after the removal of undersized segments, following the complete DSEG approach. The empty space is evaluated before it is filled with adjacent segments. A comprehensive overview of the metrics for these steps is presented in Figure 10.

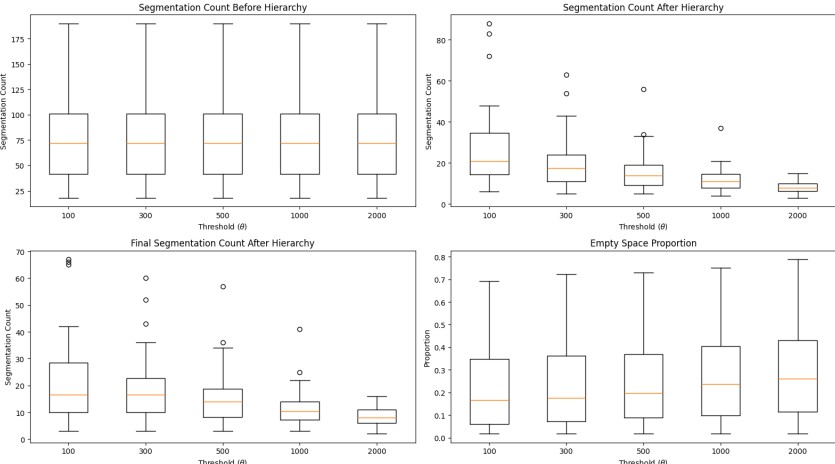

Figure 10: **Ablation study.** Here we present the interquartile range (IQR) of segmentation counts at different stages of the DSEG process (before hierarchy, after hierarchy, and final segmentation) and the proportion of empty space across various threshold values for segment size removal (denoted by $\theta$).

Higher thresholds lead to fewer segments being retained. This trend is visible in the segmentation counts before the hierarchy, after the hierarchy, and in the final segmentation. For instance, at $\theta = 100$, a higher number of segments is preserved, whereas at $\theta = 2000$, the segmentation count drops significantly due to the removal of smaller segments. Additionally, the proportion of empty space consistently increases with larger $\theta$ values. This occurs because as more small segments are removed, more unassigned or empty regions appear before being filled by adjacent segments. The increase in empty space proportion is most pronounced at higher thresholds, such as $\theta = 1000$ and $\theta = 2000$. In summary, the analysis highlights the expected trade-off between preserving smaller segments and controlling the amount of empty space. Lower thresholds result in more granular segmentation, while higher thresholds reduce the segmentation complexity at the expense

1350 of increased empty regions. Based on this trade-off, a threshold of $\theta = 500$ was selected for the
1351 experiments in this paper, as it strikes a balance between retaining meaningful segmentation detail
1352 and minimizing empty space.

### C.2 EXPLAINING WRONG CLASSIFICATION

Here, we explore how DSEG can aid in explaining a model's misclassification. Unlike the previous
analysis in Section 5.2.1, where metrics were assessed under simulations involving a model with
randomized weights (Random Model) or random predictions (Random Expl.), this case focuses on
a real misclassification by EfficientNetB4, free from external manipulation. This allows for a more
genuine examination of DSEG's ability to explain incorrect classifications under normal operating
conditions.

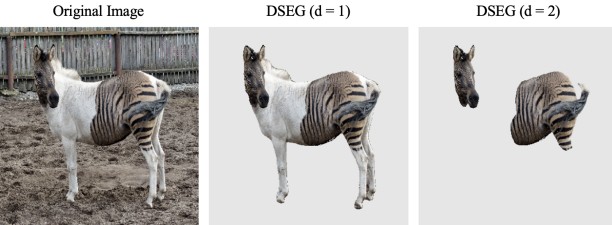

Figure 11: **Misclassification example.** The image depicts a hybrid of a horse and zebra that Ef-
ficientNetB4 classifies as a zebra with $p = 0.17$. DSEG-LIME, with a depth of one, highlights
the entire animal, offering a broad explanation. Meanwhile, DSEG at depth two pinpoints specific
zebra-like patterns that influence the model's prediction. This suggests that the model is fixating on
particular visual features associated with zebras, explaining its erroneous classification.

Figure 11 shows an image of a hybrid between a horse (sorrel) and a zebra, where EfficientNetB4
can recognize both animals but does not contain the hybrid class. We explore why EfficientNetB4
assigns the highest probability to the zebra class rather than the sorrel. Although this is not strictly a
misclassification, it simulates a similar situation and provides insight into why the model favors the
zebra label over the sorrel. This analysis helps us understand the model's decision-making process
in cases where it prioritizes specific features associated with one class over another.

### C.3 STABILITY OF EXPLANATIONS

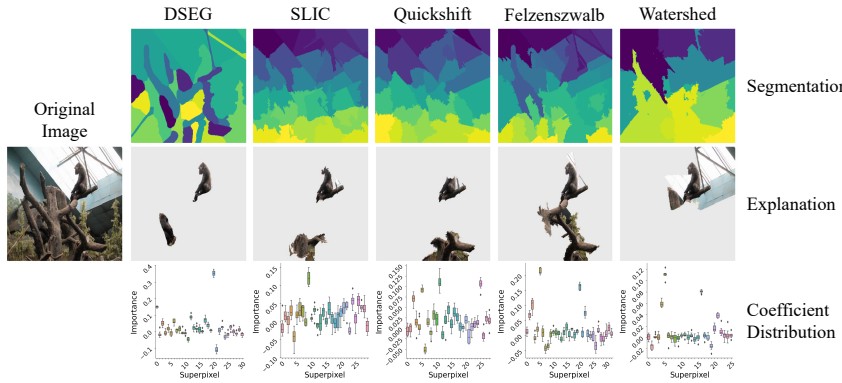

Figure 12: **Segmentation stability.** Illustrating a comparison between DSEG and other segmen-
tation techniques applied in LIME, all utilizing an identical number of samples. DSEG exhibits
greater stability compared to other segmentation techniques. Notably, the DSEG explanation dis-
tinctly highlights the segment representing a gorilla as the most definitive.

The stability of imagery explanations using LIME can be linked to the quality of feature segments, as
illustrated in Figure 12. This figure presents the segmentation maps generated by various techniques

alongside their explanations and coefficient distributions, displayed through an IQR plot over eight runs. Notably, the DSEG technique divides the image into meaningful segments; the gorilla segment, as predicted by the EfficientNetB4 model, is distinctly visible and sharply defined. In contrast, other techniques also identify the gorilla, but less distinctly, showing significant variance in their coefficient distributions. Watershed, while more stable than others, achieves this through overly broad segmentation, creating many large and a few small segments. These findings align with our quantitative evaluation and the described experimental setup.

## C.4 EXEMPLARY LIMITATION OF DSEG

The example in Figure 13 shows a complex case of a hermit crab in front of sand, which is hardly detectable. Here, SAM fails to segment the image into meaningful segments, a known issue in the community (Khani et al., 2024). In contrast, SLIC can generate segments; thus, LIME can produce an explanation that does not show a complete image.

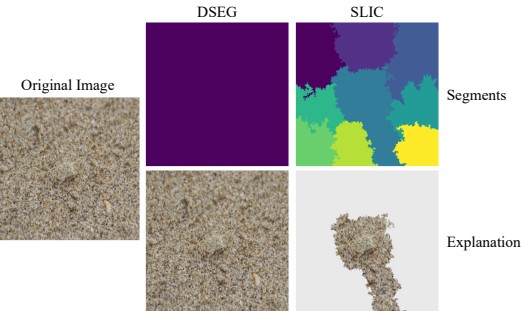

Figure 13: **DSEG fails.** Demonstrating a scenario where DSEG fails to generate meaningful features for explanations (the the whole image is one segment, in contrast to SLIC. The image shows a crab, which the model classifies as a 'hermit crab' (p = 0.17), highlighting the effectiveness of SLIC in this context compared to the limitations of DSEG.

## C.5 FEATURE ATTRIBUTION MAPS

In addition to visualizing the $n$ most essential segments for an explanation, feature attribution maps also help the explainee (the person receiving the explanation (Miller, 2019)) to get an idea of which other segments are important for interpreting the result. In these maps, the segments represent the corresponding coefficient of the surrogate model learned within LIME for the specific case. Blue segments are positively associated with the class to be explained, and red segments are negatively associated. The object representing the class is the most unique feature in all three images. We can see this particularly clearly in the image with the airplane, as the other segments have hardly any weight.

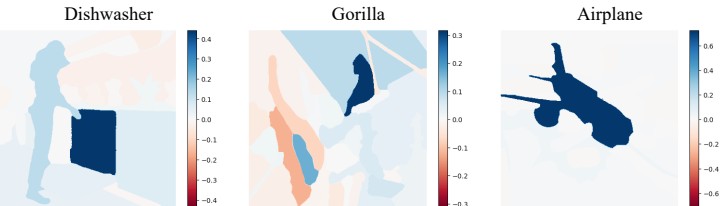

Figure 14: **DSEG attribution maps.** Representation of the feature weights of three different classified images in a feature map, with blue segments indicating positively important and red segments negatively important features in relation to the classified label. The unique blue feature indicates that the class to be explained can be recognized in all three images.

# D  DATASET AND USER STUDY

## D.1  DATASET

**Image selection.** As mentioned in Section 5.1, we selected various classes of images from the ImageNet (Deng et al., 2009) and COCO (Lin et al., 2014) dataset. Additionally, we created artificial images using the text-to-image model DALL-E (Ramesh et al., 2021) to challenge the XAI techniques when facing multiple objects. The dataset for the evaluation comprised 47 real images and three synthetic images. For the synthetic instances, the prompts 'realistic airplane at the airport', 'realistic person running in the park', and 'realistic person in the kitchen in front of a dishwasher' were used.

The object types listed in Table 16 represent the primary labels of the images used in the dataset. Each image is unique, ensuring no duplication and maximizing the diversity of animals and objects covered. The bolded types denote those that were randomly selected for qualitative evaluation. These types provide a balanced representation of the dataset and were chosen to ensure broad coverage across different categories. This selection strategy helps to avoid bias and supports a comprehensive evaluation.

Table 16: **Object families and types.** This table categorizes the images in the dataset according to their object families. The bold types indicate the classes selected for the user study, which were randomly chosen to ensure variety.

| Object Family | Type |
|---|---|
| Animals | **Ice_bear, Gorilla, Chihuahua, Husky, Horse**, Irish_terrier, Macaw, American_lobster, Kerryblue_terrier, Zebra, House_finch, American_egret, Little_blue_heron, Tabby, Black_bear, Egyptian_cat, Tusker |
| Objects | **Street_sign, Park_bench, CD_player, Banana, Projectile, Ski, Catamaran, Paper_towel, Violin, Miniskirt, Basketball, Tennis_racket, Airplane, Dishwasher, Scuba_diver**, Pier, Mountain_tent, Totem_pole, Bullet_train, Lakeside, Desk, Castle, Running_shoes, Snorkel, Digital_Watch, Church, Refrigerator, Meat_loaf, Dome, Forklift, Teddy, Mosque, Shower_curtain |

## D.2  USER STUDY

We conducted our research and user study using MTurk, intentionally selecting participants without specialized knowledge to ensure the classes represented everyday situations. Each participant received compensation of $4.50 per survey, plus an additional $2.08 handling fee charged by MTurk and $1.24 tax. The survey, designed to assess a series of pictures, takes approximately 10 to 15 minutes to complete. The sequence in which the explanations are presented to the participants was randomized to minimize bias. In our study conducted via MTurk, 59 individuals participated, along with an additional 28 people located near our research group who participated at no cost.

**Explanations.** In Figures 15a and 15b we show all 20 images from the dataset used for the qualitative evaluation. Each image is accompanied by the prediction of EfficientNetB4 and the explanations within the vanilla LIME framework with all four segmentation approaches and the DSEG variant. The segments shown in the image indicate the positive features of the explanation.

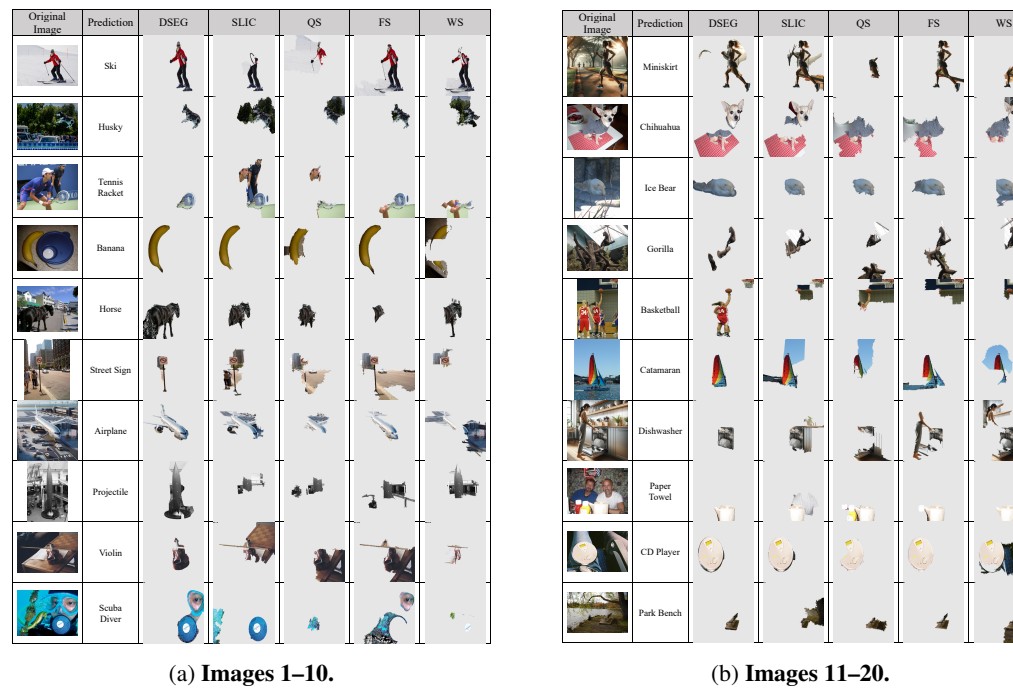

(a) **Images 1–10.**       (b) **Images 11–20.**

Figure 15: **User study data.** Examples from the evaluation datasets showing the LIME explanations alongside the original images and their corresponding predictions.

**Instruction.** Participants were tasked with the following question for each instance: 'Please arrange the provided images that best explain the concept [*model's prediction*], ranking them from 1 (least effective) to 5 (most effective).' Each instance was accompanied by DSEG, SLIC, Watershed, Quickshift, Felzenszwalb, and Watershed within the vanilla LIME framework and the hyperparameters discussed in the experimental setup. These are also the resulting explanations used in the quantitative evaluation of EfficientNetB4. Figure 16 shows an exemplary question of an instance of the user study.

**Results.** We show the cumulative maximum ratings in Figure 17a and in Figure 17b the median (in black), the interquartile range (1.5), and the mean (in red) for each segmentation technique. DSEG stands out in the absolute ratings, significantly exceeding the others. Similarly, in Figure 17b, DSEG achieves the highest rating, indicating its superior performance relative to other explanations. Therefore, while DSEG is most frequently rated as the best, it consistently ranks high even when it is not the leading explanation, as the IQR of DSEG shows. Aligned with the quantitative results in Section 5.2, the Quickshift algorithm performs the worst.

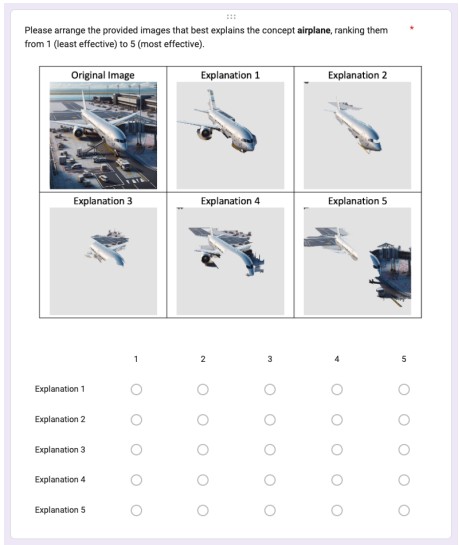

Figure 16: **Exemplary question.** The 'airplane' example is shown in the original image with its five explanations. Below the images, participants can rate the quality of the explanations accordingly.

Table 17 presents the statistical significance of the user study. Specifically, it lists the t-statistics and p-values for comparisons between DSEG (the baseline method) and other segmentation methods, namely SLIC, QS, FS, and WS. The t-statistics indicate the magnitude of difference between DSEG and each other method, with higher values representing greater differences. The corresponding p-values demonstrate the probability that these observed differences are due to random chance, with lower values indicating stronger statistical significance.

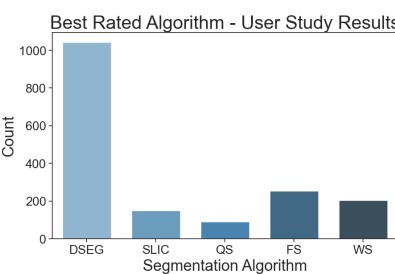

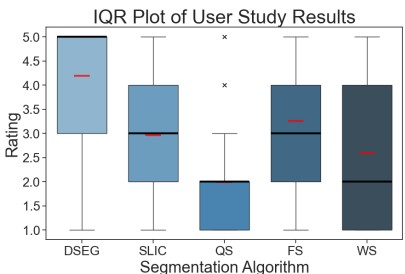

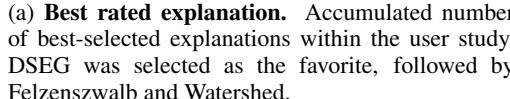

(a) **Best rated explanation.** Accumulated number of best-selected explanations within the user study. DSEG was selected as the favorite, followed by Felzenszwalb and Watershed.

(b) **IQR of explanation's ratings.** The IQR plot of the user study ratings is detailed, with the black line indicating the median and the red line representing the mean. This plot shows that DSEG received the highest ratings, while Watershed exhibited the broadest ratings distribution.

Figure 17: **User study results.** The user study ratings are visualized in two distinct figures, each employing a different form of data representation. In both visualizations, DSEG consistently outperforms the other techniques.

Table 17: **User study statistical results.** This table summarizes the statistical significance of user study results for each segmentation approach. The t-statistics and p-values indicate the comparison between DSEG and other methods. Extremely low p-values suggest strong statistical significance.

| Metric | DSEG | SLIC | QS | FS | WS |
|---|---|---|---|---|---|
| t-statistics $\uparrow$ | – | 20.01 | 49.39 | 20.89 | 33.15 |
| p-values $\downarrow$ | – | 8.0e-143 | $< 2.2$e-308 | 1.2e-86 | 3.3e-187 |

In this context, the null hypothesis ($H_0$) posits that there is no significant difference between the performance of DSEG and other segmentation methods within the LIME framework in terms of participant preferences. The alternative hypothesis ($H_A$) asserts that DSEG performs significantly better than the other segmentation methods on the dataset when evaluated based on the selection of five explanations. Given the extremely low p-values (e.g., 8.0e-143 for SLIC and ¡ 2.2e-308 for QS), we can reject the null hypothesis ($H_0$) with high confidence. The significance level of 99.9% ($\alpha = 0.001$) further supports this conclusion, as all p-values fall well below this threshold. These results indicate that the observed differences are highly unlikely to have occurred by chance and are statistically significant.

