# OpenReview forum: "DSEG-LIME: Improving Image Explanation by Hierarchical Data-Driven Segmentation"
_ICLR.cc/2025/Conference — Submitted to ICLR 2025_

### Official Review · Reviewer_1fgh · 2024-10-28

**Soundness:** 2
**Presentation:** 3
**Contribution:** 3
**Rating:** 6
**Confidence:** 3

**Summary:**

The paper describes DSEG-LIME, an improvement over the standard eXplainable AI technique LIME that uses a data-driven, AI-based, algorithm to generate the segmentation needed by LIME to assign feature attribution scores to the (super)-pixels of an image.

The paper is well presented and in the scope of the conference topics, even if it does not fully describe the impact of the presented technique in the context of learnt representations.

The main weakness (see W.2 below) is the fact that the proposed technique operates on an hierarchical segmentation with overlapping regions, which is a very complex and tricky thing to use. In fact, the LIME technique does not seem to be formulated to operate on overlapping features, as it implicitly assumes that segments are independent and form a partitioning of the pixels.
Such critical change in the LIME assumptions is not well motived, and requires additional explanations.

**Strengths:**

The paper is well motivated.
The addressed issue (data-driven XAI) is an important component missing in current state-of-the-art XAI methods.

**Weaknesses:**

**W.1**

Images of the LIME explanations apparently show the first n segments, using the very poor get_image_and_mask() method of LIME. In practice, this approach does not allow to fully understand if the method is correctly assigning large feature attribution to the right regions. A full assessment should entirely and exclusively focus on feature attribution maps, as briefly mentioned in Appendix B.5.

**W.2**

The hierarchical technique is described briefly, but does not come with a clear formulation or pseudo-code. While the intuition is somewhat clear, there are several critical details that are needed and are missing in the presentation for a full assessment.
In fact, LIME is not structurally hierarchical, and such addition would require much more than just a couple of mentions.

**W.3**

A missing analysis of DSEG-LIME is the concept of multi-scale: in principle the XAI method does not know in advance the dimension of the relevant region(s) in the input image. Your proposal account for that using some arbitrary number of iterations d of the algorithm. However, identifying how many and how large the relevant region(s) are is a very relevant problem, which does not seem to be conceptualized and approached very solidly.
A related part of this issue deals with the fact that you visualize the explanations using the first n segments instead of using feature attribution maps, which hide the actual computed relevance of the segments, and their number and areas. With this approach, the scale of the relevant regions is not well identified.

**W.4**

In Eq. (1) you use a distance function D(), which actually count the number of masked segments. Why is this a fundamental component (181)? I understand that you are using the equations of LIME (which adopt this distance function), but I do not see why this D() function should be actually relevant for your case. Do you have an explanation for why it is so critical to weight samples according to the number of masked segments? This means that the more you mask the image and focus on a specific region, the least the relevance of this sample is in the trained model.
One would guess that D() has actually very little role in the trained model, and it should actually be removed from the equation itself.

**W.5**

A critical component of LIME relates the replacement values that are used to mask the segments. There is very little analysis and reporting of that issue in the paper.

**W.6**

You report that a weakness of SAM is that it may generate overlapping segments, which makes it somewhat troublesome to deal with. The paragraph in 283-301 is not very clear about what your method does on the overlapping segments.
This is particularly critical, as the LIME method assigns feature attribution scores implicitly assuming that segments are non-overlapping partitions. If segments overlap, it is unclear what feature attribution scores means.

**W.7**

(270) What is a grid overlay parameterized by points?
Please clarify and/or add references.

**W.8**

(1384) Why are you mentioning a "NeurIPS code of Ethics" in a paper submitted to ICLR?

**Questions:**

**Q.1**
(see W.1) I would strongly suggest to use feature attribution maps to show the generated explanations among all the presented examples in the paper, instead of selecting the n most relevant segments, which is really a poor and arbitrary criteria. It does not really matter that this poor selection criteria of get_image_and_mask() was used in the 2016 LIME paper.

**Q.2**
(see W.2) Please provide a pseudo-code or a clear formulation of your hierarchical extension of LIME dealing with overlapping segments. This can be added in the Appendix.

**Q.3**
(see W.3) Is the identification of the scale of the relevant region totally left to the user which has to tune the d parameter accordingly?
Please, add a clarification to that issue.

**Q.4**
(see W.4) Can you provide intuitions why it is relevant to have the distance function D()?

**Q.6**
(see W.6) Management of overlapping regions is unclear, and require a full formulation and pseudo-code (can be added in the Appendix).
Dealing with overlapping hierarchical structures is very complex and tricky, and the paper provides very few details about how to tackle such complexity.

- Clarify what you do when you have two feature attribution scores, computed on two segments that overlap. How do you decide which one is more relevant, since the scores are not independent?
- Can you still determine a pixel-level relevance, if segments may overlap? How do you generate the images in Figure 14 when you deal with overlapping segments?
- What happens if the n topmost segments have a non-trivial overlapping region?

**Minor issues**

- (see W.5) Better clarify which replacements values you are using in the experiments (line 176 is very generic and uninformative).
- Address W.7 and W.8.

---

> ### Author Response · Authors · 2024-11-22
> **Rebuttal Part (1/2)**
>
> A revised version of the manuscript, incorporating the feedback marked with (*) in the text, will be uploaded next week. Any additional changes will be included in the final camera-ready version.
>
> We are grateful for your recognition that our paper is well-motivated and addresses an important issue in XAI. Below, we present a detailed response to each of the points raised, addressing the questions directly where applicable.
>
>
> ## Point-by-Point Response
>
> > [...] not allow to fully understand if the method is correctly assigning large feature attribution to the right regions.[...]
>
> We acknowledge that we had not previously considered this issue. Your point is well-taken (Q1), and we fully agree with your observation. We will integrate the suggested examples into our paper and explicitly highlight this issue in the main text (*), as it is indeed relevant and valuable. Additionally, we will include a quantitative assessment in the appendix (final version) to further strengthen our analysis. Thank you for bringing this to our attention!
>
> > The hierarchical technique is described briefly, but does not come with a clear formulation or pseudo-code. [...]
>
> To address your concerns, we will include additional details about the specific alterations made to LIME. As you suggested, we will integrate pseudocode (Q2) and modify the main paper to make these changes more transparent in the revised version. We believe this will resolve the issue and improve the clarity of our work(*).
>
> > [...] arbitrary number of iterations d of the algorithm. However, identifying how many and how large the relevant region(s) are is a very relevant problem, which does not seem to be conceptualized and approached very solidly [...]
>
> We appreciate your insightful comment on the multi-scale analysis of DSEG-LIME. Our goal was to conduct a basic evaluation across all methods and evaluation techniques using consistent configurations to ensure fairness and objectivity. While we recognize that fixing a parameter (d) may not fully capture the multi-scale nature of the problem, our intention was to establish a uniform evaluation framework rather than optimize for specific methods.
> To address your feedback, we will include an additional experiment with (d = 2) in Appendix A3. Furthermore, in response to concerns about attribution maps (mentioned in 1), we will incorporate this feedback while adhering to the standard procedures in LIME-related studies.
>
> > In Eq. (1) you use a distance function D(), which actually count the number of masked segments. Why is this a fundamental component (181)? [...]
>
> We confirm that we use LIME's original distance function without any modifications. However, we understand the potential confusion (Q4) caused by the way this was described. To clarify, we reference the effect of superpixel generation on this function in lines 212-214 but do not explicitly modify it. We will revise the paper to eliminate this ambiguity (*).
>
> > A critical component of LIME relates the replacement values that are used to mask the segments. [...]
>
> We take the mean pixel value of each segment in our approach. We will explicitly integrate this information into the revised version of the paper for clarity (*).
>
> > You report that a weakness of SAM is that it may generate overlapping segments, which makes it somewhat troublesome to deal with. [...]
>
> We apologize for any confusion regarding overlapping superpixels. In DSEG-LIME, we only consider **non-overlapping** superpixels. Overlap information is solely used to identify the parent segment of smaller segments. At no point does LIME assign relevance scores to the same pixel belonging to two different superpixels.
> To address unsegmented areas, we apply Equation (7) to fill these gaps with the relevance score of the nearest superpixel. This explanation also addresses the question raised in Q6. To clarify further and avoid future confusion, we will include pseudocode (Q2) and expand this explanation in the main text (*).
>
> > (270) What is a grid overlay parameterized by points? Please clarify and/or add references.
>
> The grid overlay is directly tied to SAM, which determines the number of grids used for automatic mask generation [1]. This parameter influences the number of identified segments, and we will ensure this relationship is clearly articulated in the revised paper (*).
>
> > (1384) Why are you mentioning a "NeurIPS code of Ethics" [...]?
>
> We adhere to the “NeurIPS Code of Ethics” to ensure compliance with ethical standards. We understand the confusion surrounding this reference and will revise the terminology to prevent further misinterpretation (*).

---

> > ### Author Response · Authors · 2024-11-22
> > **Rebuttal Part (2/2)**
> >
> > ## Responses to Further Questions
> >
> > Q.3: Yes, DSEG-LIME is designed to be user-friendly, allowing users to control the depth parameter (d). This is analogous to adjusting hyperparameters in vanilla LIME’s segmentation techniques, but with the added flexibility for users to dynamically choose finer or coarser segmentations for their explanations (*)
> >
> > ---
> >
> > We sincerely thank you for your constructive feedback, which will significantly improve the quality and impact of our research. While the limited rebuttal period precludes us from conducting all the necessary experiments immediately, we commit to integrating these changes in the final version of the paper. Your insights have been very valuable, and we are excited to enhance our work based on your input.
> >
> > [1]Kirillov, Alexander, et al. "Segment anything." Proceedings of the IEEE/CVF International Conference on Computer Vision. 2023.

---

> > > ### Comment · Reviewer_1fgh · 2024-11-25
> > >
> > > I believe that the rebuttal addresses the critical points, particularly Q1 and Q6, and I will revise my rating of this paper accordingly.

---

> > > > ### Author Response · Authors · 2024-11-27
> > > >
> > > > We sincerely thank you for your revised rating and are pleased that we have been able to address your concerns. In the meantime, we have uploaded a revised version of the paper that incorporates additional feedback from you, and we hope that it further strengthens the quality of our work.

---

### Official Review · Reviewer_JNcv · 2024-11-01

**Soundness:** 2
**Presentation:** 2
**Contribution:** 2
**Rating:** 3
**Confidence:** 4

**Summary:**

The paper proposes DSEG-LIME, an enhanced version of the LIME framework aimed at improving explainability for image classification tasks. By introducing a segmentation approach, DSEG-LIME leverages SAM to create human-recognizable features, enhancing the consistency and interpretability of LIME's explanations. Additionally, DSEG-LIME incorporates user-controlled segmentation granularity, allowing adjustments to the segmentation hierarchy to suit varying interpretability needs. The authors evaluate DSEG-LIME on pre-trained models with ImageNet classes, particularly focusing on applications without domain-specific knowledge.

**Strengths:**

The integration of a segmentation process aims to align explanations with human-recognizable features, potentially enhancing LIME's interpretability and making explanations more intuitive for end-users.

By allowing users to adjust segmentation granularity, the method provides flexibility, catering to various interpretability requirements across different use cases.

**Weaknesses:**

1. The novelty of this manuscript appears minimal, as it mainly combines elements from SAM and LIME without significant advancements.

2. The authors compare their method with traditional segmentation techniques, including Simple Linear Iterative Clustering (SLIC), Quickshift (QS), Felzenszwalb (FS), and Watershed. This comparison seems inadequate for assessing the effectiveness of the proposed DSEG as an XAI method. If the advantage of DSEG is due to SAM outperforming SLIC, QS, or FS, then the unique contribution of DSEG itself remains unclear. A more relevant comparison with other XAI methods is needed.

3. The authors set a threshold 𝜃=500 to filter out small masks, which is problematic, as it risks excluding areas that may contain target of interests. This thresholding strategy should be reconsidered or justified more robustly.

4. Additional visual explanations comparing DSEG with current state-of-the-art XAI methods would strengthen the manuscript by illustrating the practical benefits and effectiveness of the proposed approach.

**Questions:**

One of LIME’s main weaknesses is its instability. It is unclear if applying DSEG improves LIME’s stability or overall performance, as the results do not appear to outperform other methods. The authors should clarify this aspect and consider adding metrics like Local Fidelity from GLIME for a more comprehensive evaluation.

---

> ### Author Response · Authors · 2024-11-22
> **Rebuttal Part (1/2)**
>
> A revised version of the manuscript, incorporating the feedback marked with (*) in the text, will be uploaded next week. Any additional changes will be included in the final camera-ready version.
>
> We thank the reviewer for their thoughtful feedback and for highlighting our contribution to making LIME’s interpretations more intuitive for end-users and emphasizing its flexible approach.
>
> Below, we address the points raised in detail.
>
> ## Point-by-Point Response
>
> > [...] novelty of this manuscript [...] combines elements from SAM and LIME [...].
>
> The integration of SAM is a key motivation for using segmentation models in post-hoc explanations. DSEG-LIME acts as a framework where SAM is one possible segmentation backbone that can be utilized. In the paper, we also report statistics with DETR and SAM2. Additionally, we not only integrate SAM into the LIME framework but also construct a hierarchical tree that guides LIME in its feature importance calculations. This transforms LIME’s pipeline from a static to a dynamic approach, enabling users to explore model behavior from coarse to fine-grained concepts while offering control over the level of granularity. That said, we will revise the formulation of SAM within DSEG to shift the emphasis toward the framework itself and the overarching hierarchical concept (*).
>
> > [...] A more relevant comparison with other XAI methods is needed.
>
> We primarily compare SAM within DSEG against other mentioned segmentation techniques to evaluate how the hierarchical conceptualization built internally in DSEG compares to other approaches. Additionally, we examine how this internal representation performs at deeper levels of the hierarchy in Appendix A3. Since DSEG does not rely solely on SAM (e.g., it also utilizes SAM2 or DETR), our key contribution lies in the DSEG framework itself. This framework transforms LIME into a dynamic process by constructing a hierarchical superpixel tree, allowing to leverage various segmentation techniques as its backbone.
> In addition, incorporated an evaluation against EAC in the supplementary material, utilizing SAM as the segmentation backbone for SHAP [1]. Our main comparisons focused on DSEG against other LIME-based techniques, as highlighted in related works [2,3,4].
>
> > [...] threshold 𝜃=500 to filter out small masks, [...] strategy should be reconsidered or justified more robustly.
>
> The influence of the threshold of 500 is further investigated in Appendix B1, where we present an ablation study. While testing different values might improve DSEG-LIME’s results, we aimed to avoid extensive hyperparameter tuning in the initial quantitative study. In our view, the end user has the flexibility to decide whether smaller superpixels should be considered, much like setting hyperparameters for other segmentation techniques.
>
> > Additional visual explanations comparing DSEG with current state-of-the-art XAI methods would strengthen the manuscript[...]
>
> We will integrate more experiments with advanced techniques in the paper for camera-ready version.
>
> ---
> Answer to Question: We acknowledge that the current results may not fully demonstrate whether DSEG improves LIME’s stability. To address this, we plan to build a dataset consisting of images with more complex and diverse concepts, which are designed to challenge the stability of LIME. We will evaluate DSEG on this dataset and include these additional results in the camera-ready version.
> Furthermore, we will consider incorporating metrics such as Local Fidelity from GLIME to provide a more comprehensive evaluation of DSEG’s impact on LIME’s performance and stability.

---

> > ### Author Response · Authors · 2024-11-22
> > **Rebuttal Part (2/2)**
> >
> > We thank you for your feedback and will integrate it into a new version to enhance the quality and impact of our research at the beginning of next week. Given the limited rebuttal period, there is insufficient time to conduct all the necessary experiments, thus we will integrate them in the final version.
> > Additionally, we hope that we have effectively addressed the concerns raised and that this may lead to a reconsideration of the score. If there are any remaining concerns or additional feedback, we would be grateful to hear them. This would help us further refine our research, whether through additional experiments or by clarifying specific aspects.
> >
> > [1]Sun, Ao, et al. "Explain any concept: Segment anything meets concept-based explanation." Advances in Neural Information Processing Systems 36 (2024).
> >
> > [2]Bora, Revoti Prasad, et al. "SLICE: Stabilized LIME for Consistent Explanations for Image Classification." Proceedings of the IEEE/CVF Conference on Computer Vision and Pattern Recognition. 2024.
> >
> > [3]Meng, Han, Christian Wagner, and Isaac Triguero. "SEGAL time series classification—Stable explanations using a generative model and an adaptive weighting method for LIME." Neural Networks 176 (2024): 106345.
> >
> > [4]Bora, Revoti Prasad, et al. "SLICE: Stabilized LIME for Consistent Explanations for Image Classification." Proceedings of the IEEE/CVF Conference on Computer Vision and Pattern Recognition. 2024.

---

> > > ### Author Response · Authors · 2024-11-29
> > > **Reminder**
> > >
> > > As the review period draws to a close, we wanted to follow up to see if you have any additional questions or concerns regarding our submission. We also kindly ask if you might consider revisiting your score in light of the improvements made based on your feedback and the overall enhancements in our revised manuscript.

---

> ### Author Response · Authors · 2024-12-02
> **LIME Stability**
>
> In the meantime, we integrated your feedback by applying our DSEG framework to the SLICE [1] approach and conducted a comparative evaluation against Quickshift (QS) using hyperparameters from the SLICE repository to test stability.
>
> ### Experimental Setup
> - **Dataset:** 50 images from the main paper.
> - **Iterations:** 200 steps per image, repeated 8 times.
> - **Metrics:**
>     - **Average Rank Similarity (p = 0.3):** Focused on positive/negative segments. (Different segmentation numbers were taken into account to consider the same number of segments for computation, by min length of both)
>   - **Sign Flip Analysis:** Number of segments flipping signs during computation (Considering all segments of each technique).
>
> ### Results
>
> | Metric                          | DSEG  (pos,neg)            | QS  (pos,neg)             |
> |---------------------------------|-------------------|------------------|
> | **Average Rank Similarity (p=0.3)** | 0.969347 / 0.971652 | 0.970756 / 0.970384 |
> | **Sign Flip (mean)**            | 2.608696          | 31.288043        |
>
> ### Key Observations
> - **Similarity:** DSEG and QS exhibit comparable performance in rank similarity.
> - **Sign Flip:** DSEG significantly outperforms QS, primarily due to QS generating many segments, while DSEG inherently focuses on meaningful segments aligned with the underlying foundation model.
>
> We hope that we have clarified your question further and would be very pleased to receive a revised evaluation.
>
> [1] Bora, Revoti Prasad, et al. "SLICE: Stabilized LIME for Consistent Explanations for Image Classification." Proceedings of the IEEE/CVF Conference on Computer Vision and Pattern Recognition. 2024.

---

### Official Review · Reviewer_gLdn · 2024-11-02

**Soundness:** 2
**Presentation:** 2
**Contribution:** 2
**Rating:** 8
**Confidence:** 3

**Summary:**

The paper discusses LIME's stability problem and hypothesizes that the segmentation step is a cause for it. According to the authors, conventional segmentations are graph or clustering-based and not designed to distinguish between different objects in an image. Keeping that in mind, the authors have proposed using a transformed-based foundational model. This enables LIME's feature generation to be adapted to a novel hierarchical segmentation. Further, the authors support their claims with quantitative and qualitative studies.

**Strengths:**

1) Paper discusses about LIME stability which is a major concern to users who want to use LIME.
2) The paper performs quantative and qualitative evaluations.
3) The authors have considered a variety of models.
4) The authors integrated their proposed method, DSEG with LIME and it variants.

**Weaknesses:**

1) The literature review needs to be revised as DLIME (https://arxiv.org/abs/1906.10263) also discusses using hierarchical clustering with LIME. The authors should highlight the key differences between the proposed approach and DLIME.

2) The consistency (specially the Rep. Stability) aspect has not been investigated as there exists prior art (https://openaccess.thecvf.com/content/CVPR2024/html/Bora_SLICE_Stabilized_LIME_for_Consistent_Explanations_for_Image_Classification_CVPR_2024_paper.html) which seems to stabilize LIME explanations without investigating into the segmentation aspect. It would benefit readers if the authors explained how DSEG complements or differs from stabilization methods like SLICE and explained their rationale for focusing on segmentation rather than other stabilization techniques.

3) From Table 1 and Table 2, we can see that DSEG cannot ensure 100% consistency. Hence, I need help understanding how the other metrics (except consistency) were calculated. Does the author use the mean score of 8 runs? To improve clarity, the authors should provide more details on how they calculated metrics across multiple runs, including whether they used mean scores and how they handled any inconsistencies.

4) I need help understanding why BayLIME with non-informative prior was chosen. The authors of BayLIME showed that their method with non-informative prior was almost indistinguishable from LIME. Hence, it would be helpful for readers if the authors explained their rationale for using BayLIME with a non-informative prior and discussed how using a partial or full informative prior might impact their results.

5) The final dataset consists of 50 images, which may not be sufficient. Also, more than the number of runs (i.e., 8) may be needed for the Rep. Stability test. Hence, I would like the authors to justify their choice of dataset size and number of runs by referencing similar studies or discussing statistical power considerations. There is mention of t-statistic and p-value for user-study results, but it would be helpful for readers if the authors explained why they did not use the same for the quantitative evaluation. Further, the authors should also describe their choice of the test and the alternative hypothesis used for ease of readers and complement it with effect size.

6) For Consistency, SLIC is at par (or very close) with DSEG in many situations (Table 1 and Table 2), (Table 6 and Table 7). It would be helpful to readers if authors explain why SLIC performs similarly to DSEG in terms of consistency in some situations and discuss potential factors that might contribute to this observation.

**Questions:**

I request the authors to look at the weaknesses and provide their comments. I think the paper addresses an important aspect of LIME and the proposed approach has merits, but some aspects need polishing. I would like to know the authors' comments regarding my questions.

---

> ### Author Response · Authors · 2024-11-22
> **Rebuttal Part (1/2)**
>
> A revised version of the manuscript, incorporating the feedback marked with (*) in the text, will be uploaded next week. Any additional changes will be included in the final camera-ready version.
>
> We thank the reviewer for acknowledging the importance of addressing LIME’s stability concerns and evaluating it rigorously through quantitative and qualitative methods. Additionally, integrating our proposed method, DSEG, with LIME and its variants demonstrates the versatility of our approach across a variety of models.
>
> ## Point-by-Point Response
>
> >  [...] revised as DLIME [...] also discusses using hierarchical clustering with LIME.[...]
>
> DLIME is indeed a relevant work that we will integrate into the manuscript accordingly (*). However, in contrast to DLIME, DSEG-LIME operates on the image modality. In addition, our hierarchy directly relates to the composition of image segments, providing LIME with a dynamic and adaptable approach. In contrast, DLIME employs agglomerative hierarchical clustering to group training data, and focusing on tabular datasets.
> The rational by focusing on segmentation for stability is finding the model's decision boundry closer than other segmentation methods would do. Compared to e.g., Slice handling stability with entropy calculation.
>
> > [...] seems to stabilize LIME explanations without investigating into the segmentation aspect. It would benefit readers [...] complements or differs from stabilization methods [...]
>
> We developed DSEG-LIME because many LIME-based approaches, including methods like SLICE, do not specifically investigate the stability of explanations through the segmentation process itself. This is a critical aspect that we believe requires closer examination. Therefore, we created DSEG-LIME, building on the principles outlined in [1], to provide a more detailed and systematic analysis of how segmentation impacts the stability of LIME explanations. However, we understand this point and will include a comparison of the methods (*).
>
> > [...] mean score of 8 runs? To improve clarity, the authors should provide more details on how they calculated metrics across multiple runs, including whether they used mean scores and how they handled any inconsistencies.
>
> For other metrics except of stability, we present results based on one run, thus not taking the mean, due to runtime of all results for all models and LIME combinations.
>
> >[...] why BayLIME with non-informative prior was chosen. [...]
>
> We fully agree that BayLIME with non-informative prior is nearly indistinguishable. Our rationale behind this evaluation was to stick to a setting, where we do not build any priors into the process. Since we also integrate DSEG within BayLIME, there was no disadvantage regarding this method and ensured an equal comparison against other LIME backbones.
>
>
> >[...] dataset consists of 50 images, which may not be sufficient. Also, more than the number of runs (i.e., 8) may be needed for the Rep. Stability test. [...]
>
> We use 50 images, like related approaches [2]. After consideration and discussion with other authors, we acknolwedge that evaluating more than 50 images, and increasing it with other works to 100 images makes our approach more robust. Regarding qualitative evaluation, we stick to random 20 images to have a greater audience of evaluating the explanations than having only a few participant evaluating the whole 50 (or in the future 100) images.
> For the user study, we used t-statistics and p-values to evaluate user performance and preferences, as this setting involved comparing groups and drawing conclusions about their significance. However, these metrics were not applied in the quantitative evaluation because the focus there was on measuring performance metrics rather than testing differences between groups, but we will add this information with the increased dataset.
> To clarify, our alternative hypothesis is as follows:
> *DSEG performs significantly better than other segmentation methods within the LIME framework in terms of preference by the participants, when applied to the dataset with the selection of five explanations.* (*)

---

> > ### Author Response · Authors · 2024-11-22
> > **Rebuttal Part (2/2)**
> >
> > >[...] SLIC is at par (or very close) with DSEG [...] explain why SLIC performs similarly to DSEG in terms of consistency [...]
> >
> > Thank you for your insightful comments. In our opinion, the reason for the observed similarities in performance between SLIC and DSEG within the LIME framework lies in the inherent reliance of LIME on localized perturbations and the subsequent surrogate model. The surrogate model’s influence may sometimes overshadow in segmentation granularity between SLIC and DSEG, particularly in scenarios where certain image characteristics—such as low texture complexity, uniform colors, or strong edge definition—reduce the advantage of SAM-based segmentation.
> > Moreover, the comparable number of superpixels ensures that no segmentation technique is overly advantaged, as the compactness size was chosen to maintain fairness. Additionally, the choice of pixel replacement value, where we use the mean pixel value for superpixel replacement, could further impact the observed performance differences.
> > We will incorporate this discussion in the refined version of the manuscript (*) to provide a clearer explanation of these factors and their potential effects.
> >
> > ---
> > We would like to kindly ask for further clarification regarding the handling of inconsistencies, as we are unsure about the specific aspect of the third question.
> >
> > We sincerely thank the reviewer for the valuable questions and suggestions, which we plan to incorporate to enhance the presentation of our research. Additionally, we hope that we have effectively addressed the concerns raised and that this may lead to a reconsideration of the score. If there are any remaining concerns or additional feedback, we would be grateful to hear them. This would help us further refine our research, whether through additional experiments or by clarifying specific aspects.
> >
> > [1]Ng, Chung Hou, Hussain Sadiq Abuwala, and Chern Hong Lim. "Towards more stable lime for explainable ai." 2022 International Symposium on Intelligent Signal Processing and Communication Systems (ISPACS). IEEE, 2022.
> > [2]Bora, Revoti Prasad, et al. "SLICE: Stabilized LIME for Consistent Explanations for Image Classification." Proceedings of the IEEE/CVF Conference on Computer Vision and Pattern Recognition. 2024.

---

> ### Comment · Reviewer_gLdn · 2024-11-26
> **Regarding Statistical Tests**
>
> **1) Author's comments:**
> "We developed DSEG-LIME because many LIME-based approaches, including methods like SLICE, do not specifically investigate the stability of explanations through the segmentation process itself. This is a critical aspect that we believe requires closer examination. Therefore, we created DSEG-LIME, building on the principles outlined in [1], to provide a more detailed and systematic analysis of how segmentation impacts the stability of LIME explanations. However, we understand this point and will include a comparison of the methods (*)."
>
> **Further Question:**
> SLICE stabilizes LIME explanations without investigating the segmentation aspect, but I did not find any reason why it cannot be compared with the proposed method. I think the authors need not run additional experiments for this but can clarify their reason for not comparing with SLICE or other such methods.
>
> **2) Author's comments:**
> "For the user study, we used t-statistics and p-values to evaluate user performance and preferences, as this setting involved comparing groups and drawing conclusions about their significance."
>
> **Further Question:**
> What is the rationale behind the choice of a parametric test like the t-test? How did the authors verify that the data follows the assumptions made by the test?
>
> **3) Author's comments:**
> "However, these metrics were not applied in the quantitative evaluation because the focus there was on measuring performance metrics rather than testing differences between groups."
>
> **Further Question:**
> I request the authors to explain this statement.
>
> **4) Author's comments:**
> "We use 50 images, like related approaches [2]. After consideration and discussion with other authors, we acknowledge that evaluating more than 50 images, and increasing it with other works to 100 images makes our approach more robust."
>
> **Further Question:**
> My concern was not just the number of images but also the number of runs. I request the authors to explain how running the proposed method only 8 times can help in robust estimation of Rep. Stability.

---

> > ### Author Response · Authors · 2024-11-26
> > **Anwer to Statistical Tests**
> >
> > We appreciate the thoughtful follow-up questions and address them in detail:
> >
> > > SLICE stabilizes LIME explanations without investigating the segmentation aspect, but I did not find any reason why it cannot be compared with the proposed method. I think the authors need not run additional experiments for this but can clarify their reason for not comparing with SLICE or other such methods..
> >
> > We did not intend to overlook or exclude SLICE and acknowledge the importance of comparing it with the proposed method. For completeness, we will integrate SLICE into the camera-ready version to provide a more comprehensive evaluation. Our goal is to ensure that the work is robust and well-situated in the context of related techniques. Additionally, we already include comparisons with similar methods aimed at improving stability, such as SLIME, GLIME, and BayLIME.
> >
> > >  What is the rationale behind the choice of a parametric test like the t-test? How did the authors verify that the data follows the assumptions made by the test?
> >
> > Regarding the assumptions of the t-test:
> >
> > 1. Normality: The responses in the user study were measured on continuous scales (e.g., ratings on a 1-to-5 scale), which are more likely to approximate normal distributions. However, to address potential concerns about violations of this assumption, we conducted the non-parametric Wilcoxon signed-rank test as a robustness check. Non-parametric tests do not assume normality, providing a complementary perspective to the paired t-tests. The results are as follows:
> >
> >     - Comparison DSEG vs SLIC: Statistic = 278,425.5, p-value = 1.28e-118
> >     - Comparison DSEG vs QS: Statistic = 122,147.0, p-value = 2.04e-205
> >     - Comparison DSEG vs FS: Statistic = 384,093.0, p-value = 1.28e-73
> >     - Comparison DSEG vs WS: Statistic = 224,598.5, p-value = 6.17e-145
> >
> >     These extremely small p-values align with the t-test results, strengthening our findings and suggesting that the results are robust across parametric and non-parametric approaches.
> >
> > 2. Independence: We ensured that tasks were completed independently, without collaboration or influence between participants. Participants were randomly assigned tasks using MTurk to further reduce potential biases.
> >
> > 3. Homogeneity of Variance: The task difficulty, environmental conditions, and other factors affecting responses were kept consistent across all individual studies to ensure comparable variance across groups.
> >
> > The choice of the t-test was grounded in the nature of the data and its ability to provide powerful statistical inferences. However, we supplemented our analysis with the Wilcoxon signed-rank test to account for any concerns about the parametric assumptions. Both methods lead to consistent conclusions, bolstering the validity of our results.
> >
> > > However, these metrics were not applied in the quantitative evaluation because the focus there was on measuring performance metrics rather than testing differences between groups.
> >
> > Our focus was to measure quantitative metrics consistent with prior works, particularly those outlined in the systematic review on evaluating explainable AI [1]. The primary goal was to follow established practices rather than test group differences in this specific context. We acknowledge this limitation and will address it by incorporating further clarity and additional statistical analyses where relevant in the final version.
> >
> > > My concern was not just the number of images but also the number of runs. I request the authors to explain how running the proposed method only 8 times can help in robust estimation of Rep. Stability.
> >
> > Apologies for misunderstanding your question earlier. Our rationale for running the method 8 times was based on prior qualitative assessments, which indicated that this number adequately captures variance in explanations, particularly in attribution scores of features, while balancing computational resource constraints. However, if requested, we are open to increasing the number of runs in the final version to make the robustness of our estimates more rigorous and statistically justified.
> >
> >
> > [1]Nauta, Meike, et al. "From anecdotal evidence to quantitative evaluation methods: A systematic review on evaluating explainable ai." ACM Computing Surveys 55.13s (2023): 1-42.

---

> > > ### Author Response · Authors · 2024-11-30
> > > **DSEG in SLICE**
> > >
> > > In the meantime, we integrated your feedback by applying our DSEG framework into the SLICE approach and conducted a comparative evaluation against Quickshift (QS) using hyperparameters from the SLICE repository.
> > >
> > > ### Experimental Setup
> > > - **Dataset:** 50 images from the main paper.
> > > - **Iterations:** 200 steps per image, repeated 8 times.
> > > - **Metrics:**
> > >     - **Average Rank Similarity (p = 0.3):** Focused on positive/negative segments. (Different segmentation numbers were taken into account to consider the same number of segments for computation, by min length of both)
> > >   - **Sign Flip Analysis:** Number of segments flipping signs during computation (Considering all segments of each technique).
> > >
> > > ### Results
> > >
> > > | Metric                          | DSEG  (pos,neg)            | QS  (pos,neg)             |
> > > |---------------------------------|-------------------|------------------|
> > > | **Average Rank Similarity (p=0.3)** | 0.969347 / 0.971652 | 0.970756 / 0.970384 |
> > > | **Sign Flip (mean)**            | 2.608696          | 31.288043        |
> > >
> > > ### Key Observations
> > > - **Similarity:** DSEG and QS exhibit comparable performance in rank similarity.
> > > - **Sign Flip:** DSEG significantly outperforms QS, primarily due to QS generating many segments, while DSEG inherently focuses on meaningful segments aligned with the underlying foundation model.
> > >
> > > These findings highlight that DSEG requires less hyperparameter tuning and offers more robust segmentation compared to alternative approaches. We hope this additional analysis further supports our research.

---

> > > > ### Comment · Reviewer_gLdn · 2024-12-02
> > > >
> > > > I thank the authors for their detailed explanations. I will update my ratings.

---

> > > > > ### Author Response · Authors · 2024-12-03
> > > > >
> > > > > Thank you sincerely for updating your rating; we’re glad we could successfully address your points.

---

### Official Review · Reviewer_ieVc · 2024-11-04

**Soundness:** 2
**Presentation:** 2
**Contribution:** 2
**Rating:** 3
**Confidence:** 4

**Summary:**

The paper introduces a new approach for explaining image-classifier models by using hierarchical segmentation, which provides explanations at multiple levels of granularity. This method leverages foundation models to extract concepts that are more aligned with human perception, creating explanations that are easier to understand and interpret. Unlike traditional LIME, which often faces challenges in achieving both human interpretability and faithful representations of model behavior, this approach enhances explainability by structuring explanations in a way that mirrors human visual processing.

**Strengths:**

- It is an interesting approach with real problems to address.
- It effectively leverages SAM’s capabilities for hierarchical segmentation, enhancing explainability by capturing meaningful image segments.
- The user study demonstrates that the extracted concepts are recognizable and understandable for humans.
- Empirical results support the effectiveness of the proposed method in generating more interpretable and faithful explanations compared to traditional techniques.

**Weaknesses:**

- Does SAM consider an image as an input or extracted features from a classifier as an input, figure 2 points at extracted features, while equation 3 points at input image?
- In the spirit of LIME the surrogate model is trained to mimic the behaviour of the given classifier, but in this scenario, the surrogate model not only considers true model, but also SAM masks with input.
    - Given the above statement, the conducted human evaluation states SAM’s capabilities to capture human-recognisable features and not the classifiers perception of these concepts, I would like to know authors thoughts on this?
    - Additionally, DSEG offsets computational cost of extracting explanations as compared to LIME
- I would like to know authors view-point on explaining one deep-learning model using another much deeper more complex model?
- The output from the foundation model isn’t always human-interpretable, highlighting the need for more quantitative studies to evaluate interpretability in greater depth.

**Questions:**

see weaknesses

---

> ### Author Response · Authors · 2024-11-22
> **Rebuttal Part(1/2)**
>
> A revised version of the manuscript, incorporating the feedback marked with (*) in the text, will be uploaded next week. Any additional changes will be included in the final camera-ready version.
>
> We appreciate the reviewer’s feedback and for recognizing the strengths of our approach, including its ability to tackle real-world interpretability challenges and enhance explainability through SAM’s hierarchical segmentation. The user study and empirical results further highlight the method’s effectiveness in generating interpretable and faithful explanations.
>
> ## Point-by-Point Response
>
> > Does SAM consider an image as an input or extracted features from a classifier as an input, figure 2 points at extracted features, while equation 3 points at input image?
>
> We agree with your observation regarding the input image processing in Figure 2. We will revise the figure to better reflect the workflow and input structure to ensure clarity and accuracy (*).
>
> > Given the above statement, the conducted human evaluation states SAM’s capabilities to capture human-recognisable features and not the classifiers perception of these concepts, I would like to know authors thoughts on this?
>
> We integrate SAM and other segmentation models in a manner analogous to techniques like SLIC. Specifically, the final masks used by LIME (after removing empty spaces) correspond to superpixels formatted similarly to those generated by SLIC. This ensures consistency in the pipeline and comparability across methods. We understand the confusion here and will clarify it in Section (5.3) (*).
>
> In our qualitative evaluation, we compare explanations generated using different backbones and assess human subjects' preferences for specific classifications. The results demonstrate that SAM’s segmentation tendencies—which align with human-recognizable objects and classes—produce explanations that are preferred by users. This outcome underscores SAM’s ability to generate features that resonate with human intuition, enhancing the interpretability of the explanations.
>
> > Additionally, DSEG offsets computational cost of extracting explanations as compared to LIME
>
> Regarding computation time, we acknowledge the trade-off but emphasize that the improved quality of the results, both qualitatively and quantitatively, justifies and offsets the associated costs. The higher interpretability and fidelity of explanations lead to tangible benefits, including better alignment with human understanding, enhanced usability, and greater trust in the model outputs. We believe, these benefits validate the additional computational investment.
>
> > I would like to know authors view-point on explaining one deep-learning model using another much deeper more complex model?
>
> We recognize the concern regarding the interpretability of deep models when utilizing even larger foundation models for their explanation. While this shifts the problem to a more complex model, our work, along with other studies, demonstrates that large foundation models offer unique advantages in interpreting smaller models [1, 2, 3,4].
> These examples show that leveraging the hierarchical and semantic understanding of large models can meaningfully address interpretability challenges. Furthermore, since our goal is to explain the local behavior of models like EfficientNet around a specific instance, the trustworthiness of the explanation remains intact
>
> > The output from the foundation model isn’t always human-interpretable, highlighting the need for more quantitative studies to evaluate interpretability in greater depth.
>
> We acknowledge that the outputs of foundation models can sometimes lack direct human interpretability. This reinforces the importance of robust evaluation methodologies that go beyond subjective human assessments. In response, we conducted a comprehensive quantitative evaluation to complement qualitative assessments. This ensures that our findings are not only based on subjective human preferences but are also supported by objective, data-driven insights.
>
> However, we agree that evaluating segmentation interpretability requires further exploration. Future work should investigate the qualitative segmentation performance of such models to deepen our understanding of their interpretability capabilities.

---

> > ### Author Response · Authors · 2024-11-22
> > **Rebuttal Part (2/2)**
> >
> > We thank the reviewer for their questions and feedback. As stated above, we are going to integrate your remarks in a newer version we upload next week. This will allow us to refine our approach, incorporate the suggestions, and polish the paper to ensure clarity and completeness in addressing the identified points.
> >
> > Additionally, we hope that we have effectively addressed the concerns raised and that this may lead to a reconsideration of the score. If there are any remaining concerns or additional feedback, we would be grateful to hear them. This would help us further refine our research, whether through additional experiments or by clarifying specific aspects.
> >
> > #### Sources:
> > [1] Kroeger, Nicholas, et al. "Are Large Language Models Post Hoc Explainers?." arXiv preprint arXiv:2310.05797 (2023)
> >
> > [2]Sun, Ao, et al. "Explain any concept: Segment anything meets concept-based explanation." Advances in Neural Information Processing Systems 36 (2024).
> >
> > [3]Yang, Yue, et al. "Language in a bottle: Language model guided concept bottlenecks for interpretable image classification." Proceedings of the IEEE/CVF Conference on Computer Vision and Pattern Recognition. 2023.
> >
> > [4]Kweon, Hyeokjun, and Kuk-Jin Yoon. "From SAM to CAMs: Exploring Segment Anything Model for Weakly Supervised Semantic Segmentation." Proceedings of the IEEE/CVF Conference on Computer Vision and Pattern Recognition. 2024.

---

> > > ### Author Response · Authors · 2024-11-29
> > > **Reminder**
> > >
> > > As the review period draws to a close, we wanted to follow up to see if you have any additional questions or concerns regarding our submission. We also kindly ask if you might consider revisiting your score in light of the improvements made based on your feedback and the overall enhancements in our revised manuscript.

---

### Author Response · Authors · 2024-11-22
**Rebuttal Summary**

We sincerely thank the reviewers for their constructive feedback and for recognizing the strengths of our method. In particular, we appreciate the reviewers [ieVc, gLdn, JNcv, 1fgh] highlighting the motivation and relevance of our approach. Notably, the reviewers underscored our method’s advantage of enabling user-steered explanation granularity [JNcv, ieVc], as well as the quantitative and qualitative assessment of DSEG-LIME [ieVc, gLdn].

Below, we summarize the main concerns raised by the reviewers and detail how we have addressed each one. We have provided individualized responses to all questions directly to each reviewer. Additionally, we will upload a revised version of the manuscript next week, incorporating feedback marked with (*). Further adjustments, which are constrained by time, will be included in the camera-ready version.

- **Segmentation Model Integration** [ieVc, JNcv, 1fgh, gLdn]:
To address the feedback, we will revise the related work section to better differentiate DSEG-LIME from other LIME-related techniques. Furthermore, we will enhance the introduction of DSEG-LIME to clearly present it as a framework capable of integrating any segmentation model for superpixel generation and organizing them into a hierarchical tree to enable a dynamic variant of LIME. To clarify this, we will make adjustments to Figure 2, add pseudocode in the appendix, and emphasize the framework itself rather than focusing predominantly on SAM. These changes aim to provide a clearer understanding of our contribution as a flexible and adaptable framework.

- **Stability of DSEG-LIME** [JNcv, 1fgh, gLdn]:
We acknowledge the reviewers’ concerns regarding the stability of DSEG-LIME. We have addressed these concerns as thoroughly as possible in our current responses. Additionally, we plan to include further experiments in the camera-ready version to provide deeper insights into stability and robustness. These experiments will specifically target scenarios designed to test the stability of LIME under varying conditions and will further validate the benefits of our approach.

---

### Meta-Review · Area_Chair_hMS9 · 2024-12-17

**Metareview:**

This submission initially had divergent ratings. The major concerns raised were:

1. the surrogate model of LIME uses both the SAM masks as input -- does the human evaluation then reveal SAM's capability of capturing human recognizable features rather than the classifiers? [ieVc]
2. Explaining one deep-learning model using another much deeper more complex model [ieVc]
3. the output of the foundation model isn't always human interpretable, so more quantitative analysis is required. [ieVc]
4. Evaluation dataset is too small (50 images) [gLdn]
5. Need more information about consistency and alternative stabilization methods [gLdn]
6. Limited novelty - combining SAM and LIME [JNcv]
7. the main advantage of DSEG is due to using SAM over SLIC or other traditional segmentation methods, which is not convincing in terms of unique contribution for DSEG itself. [JNcv]
8. threshold of 500 will filter out small masks of small targets [JNcv]
9. missing comparison with SOTA AI methods. [JNcv]
10. missing analysis of DSEG-LIME from multi-scale perspective, and other descriptions (pseudo-code) [1fgh]
11. needs better visualization using feature attribution maps [1fgh]
12. how to manage overlapping regions [1fgh]

The authors wrote a response. While some reviewers were satisfied, others were not. In particular, points 1, 2, 4, 6, 7, 9, which are about novelty, limited experiments, lack of comparison with SOTA, and motivation of using a larger deep learning model to explain a smaller one, were not addressed well.

The AC agrees with these concerns, and thus recommends reject.

**Additional Comments On Reviewer Discussion:**

see above

---

### Decision · Program_Chairs · 2025-01-22

Reject